# SyncLipMAE: Contrastive Masked Pretraining for Audio–Visual Talking-Face Representations

## Abstract

We introduce SyncLipMAE, a self-supervised pretraining framework for talking-face video that learns synchronization-aware and transferable facial dynamics from unlabeled audio–visual streams. Our approach couples masked visual modeling with cross-modal contrastive alignment and employs three per-frame prompt tokens that explicitly encode the essential factors of a talking-face frame—identity, vocal motion (speech-synchronized facial dynamics), and ambient motion (audio-agnostic movements such as blinks and head pose). The contrastive objective uses time-aligned vocal-motion and audio tokens as positives and misaligned pairs as negatives, driving both modalities into a shared embedding space and yielding token-level audio–visual stream synchronization. After pretraining, the aligned audio tokens together with the visual prompt tokens (identity, vocal motion, ambient motion) form a unified interface for four disparate downstream settings: (i) audio–visual stream synchronization; (ii) facial emotion and head/face action recognition; (iii) visual speech recognition; and (iv) visual dubbing, for which we, for the first time, enable indistinguishable audio- or video-driven control within a single model. Across four task families that require distinct capabilities, SyncLipMAE achieves state-of-the-art results, underscoring the effectiveness of synchronization-aware, factorized self-supervised pretraining.

## 1 Introduction

Talking-face video underpins human–computer interaction, accessibility, content creation, and post-production. Practical systems must jointly reason about linguistic content, audio–visual timing, video editing to match a target speech track or reference motion, and facial expressions/actions. Classic synchronization pipelines such as SyncNet (Chung & Zisserman, 2016) and subsequent Transformer-based variants (e.g., VocaLiST (Kadandale et al., 2022)) focus on detecting in-/out-of-sync pairs or removing stream lags, while recent audio–visual pretraining methods (e.g., AV-HuBERT (Shi et al., 2022)) have shown strong transfer to lip reading and AV-SR. Yet, these advances do not directly yield a *token-level*, stream-synchronised representation that cleanly separates identity, speech-synchronised mouth motion, and other facial dynamics—capabilities that are essential for both analysis and controllable generation.

Most prior work tackles one facet at a time. Visual or video masked modeling (MAE (He et al., 2022)/VideoMAE (Tong et al., 2022)) excels at reconstruction but is modality-specific and does not enforce audio–visual alignment; contrastive AV learning improves correspondence but is typically framed as binary sync classification rather than a shared token space; and dubbing systems (e.g., Wav2Lip (Prajwal et al., 2020), LatentSync (Li et al., 2024), MuseTalk (Zhang et al., 2024b)) optimise for generation quality and lip accuracy but are not designed to provide a unified interface that can be driven interchangeably by audio or reference motion within a single model. As a result, existing solutions often rely on wider temporal windows or task-specific heads, and they lack a factorised, alignment-aware representation that transfers seamlessly across synchronisation, understanding, and dubbing.

We introduce SyncLipMAE, a self-supervised pretraining framework tailored to talking-face video. SyncLipMAE learns three per-frame prompt tokens—**identity**, **vocal motion** (speech-synchronised

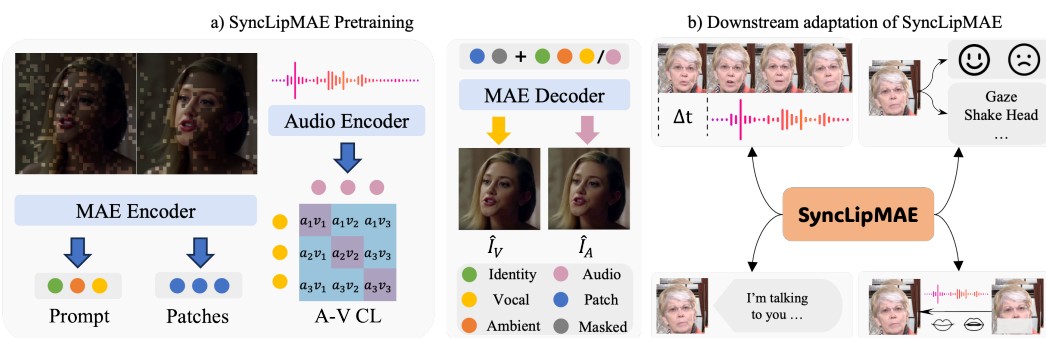

Figure 1: Panel (a) schematically illustrates the core components of SyncLipMAE and the computational pipeline used during pretraining, while panel (b) shows its adaptation to downstream tasks.

dynamics), and **ambient motion** (audio-agnostic movements)—and couples masked visual reconstruction with a cross-modal contrastive objective that aligns vocal-motion tokens to temporally matched audio tokens in a shared embedding space. A two-view masking strategy (uniform vs. face-preserving) sources patch context and factor prompts, and a simple identity-shuffle regulariser further purifies the identity token. Decoding is performed in two passes per frame (video- and audio-conditioned) with shared weights, which provides symmetric supervision that tightens audio–visual stream synchronization at the token level. This factorised, aligned interface directly supports diverse downstream uses without task-specific architecture changes.

Beyond analysis, SyncLipMAE enables *unified* visual dubbing: using a WanVACE-style (Jiang et al., 2025) DiT backbone for masked video inpainting, we inject either aligned audio tokens or vocal-motion tokens *after* patch embedding via a lightweight AudioPack, so the same model admits no-difference audio- or video-driven control. This preserves identity and pose while re-synchronising the mouth region, and it naturally benefits from the shared token space learned during pretraining.

**Contributions.** (i) We propose a synchronization-aware, factorised pretraining scheme that yields three prompt tokens (identity, vocal motion, ambient motion) and a shared audio–visual token space. (ii) We introduce a two-view masking design and identity-shuffle regularisation tailored to talking faces. (iii) We show a unified dubbing interface that *for the first time* enables no-difference audio- or video-driven control within a single model by injecting aligned control tokens into a WanVACE backbone. (iv) We demonstrate state-of-the-art performance across four distinct tasks—audio–visual stream synchronization, facial expression/action understanding, visual speech recognition, and visual dubbing—highlighting the effectiveness and generality of the approach.

## 2 RELATED WORK

**Audio–visual synchronisation and correspondence.** Audio–visual learning is often formulated as correspondence or temporal alignment. L3-Net Arandjelovic & Zisserman (2017) learns audio–visual correspondence from unlabelled video, while AVTS Morgado et al. (2020) casts synchronisation as in-time vs. out-of-time discrimination within a clip. For faces, SyncNet Chung & Zisserman (2016) introduces a two-stream embedding for lip–audio alignment and lag estimation, and Transformer-based models such as VocaLiST Kadandale et al. (2022) improve robustness across speech and singing with longer-range context.

**Masked modelling and contrastive learning at scale.** Contrastive pretraining aligns paired modalities at scale (e.g., CLIP (Radford et al., 2021)), while masked autoencoders (MAE (He et al., 2022), VideoMAE (Tong et al., 2022)) show that reconstructing heavily masked inputs yields strong visual and video representations. These two paradigms are now standard building blocks for multimodal pretraining.

**Facial video representation pretraining.** MARLIN (Cai et al., 2023) applies masked autoencoding to facial videos, using facial-region-guided masking to obtain a universal face encoder transferable to

expression recognition, deepfake detection, and related tasks. Facial Region Awareness (FRA) Gao & Patras (2024) further discovers facial regions via learned heatmaps and enforces consistency between global and local features, improving transfer across facial analysis benchmarks; subsequent variants adapt MAE-style pretraining to dynamic facial expression recognition under limited labels (Sun et al., 2023). These methods focus on reconstruction-only objectives within the facial domain, whereas we couple MAE with audio–visual contrast to align speech-driven dynamics in talking-face video and explicitly separate identity and motion factors.

**Visual speech recognition and audio–visual ASR.** End-to-end lipreading, large-scale audio–visual pretraining, and automatic recipe design have steadily improved VSR/AVSR. AV-HuBERT learns audio–visual speech representations from unlabelled video, and Auto-AVSR (Ma et al., 2023) automates model and loss selection for VSR/AVSR pipelines. ES3 (Zhang et al., 2024a) decomposes audio–visual self-supervised learning into shared, modality-specific, and synergistic components and introduces an evolving Siamese training strategy, yielding strong low-resource VSR/AVSR. Our work is complementary: we also learn joint audio–visual speech–face representations, but optimise them for speech-driven facial dynamics and dubbing/synchronisation rather than word recognition.

**Talking-face generation and lip-sync synthesis.** Audio-driven talking-head synthesis has progressed from GAN-based pipelines with explicit sync critics (e.g., Wav2Lip (Prajwal et al., 2020)) to latent/image-space diffusion (e.g., LatentSync (Li et al., 2024)) and efficient diffusion heads such as MuseTalk (Zhang et al., 2024b). AV-HuBERT-based lip-sync experts Yaman et al. (2024) instead treat a pretrained audio–visual speech model as a frozen teacher, using its features to define lip-sync losses and evaluation metrics. In parallel, foundation video generators and conditional DiT frameworks (DiT (Peebles & Xie, 2023), WAN (Wan et al., 2025), VACE (Jiang et al., 2025)) provide scalable backbones and conditioning interfaces that we leverage in our dubbing setup. Unlike Yaman et al. (2024), we pretrain a dedicated face encoder jointly with audio rather than relying solely on a frozen AV speech expert.

## 3 SyncLipMAE: Pretraining Objective and Architecture

### 3.1 Overview and Design Goals

SyncLipMAE aims to learn synchronization-aware talking-face representations from unlabeled audio–visual streams. As illustrated in figure 1(a), we combine MAE-style masked video modeling with a CLIP-style contrastive loss that pulls time-aligned *vocal-motion* and *audio* tokens together and pushes temporally misaligned pairs apart, yielding a shared token-level audio–visual space (§3.3). For each frame, a ViT encoder outputs three prompt tokens for identity $\mathbf{z}^{\text{id}}$, vocal motion $\mathbf{z}^{\text{voc}}$, and ambient motion $\mathbf{z}^{\text{amb}}$, and the decoder reconstructs the frame via cross-attention to these prompts.

**Inputs and Notation.** Let $\mathbf{x}_{1:T} \in \mathbb{R}^{T \times C \times H \times W}$ be the video frames and $\mathbf{a} \in \mathbb{R}^N$ a mono waveform at sampling rate $f_s$. Each frame is patchified into $N_p$ patches and embedded as $D$-dimensional tokens. For MAE-style masking, let $\mathcal{V}_t \subset \{1, \ldots, N_p\}$ be the visible indices at time $t$ and $\mathcal{M}_t$ the masked ones, with $|\mathcal{V}_t| = N_{\text{vis}}$ and $\mathcal{V}_t \cup \mathcal{M}_t = \{1, \ldots, N_p\}$; the visible patch tokens are $\mathbf{P}_t^{\text{vis}} \in \mathbb{R}^{N_{\text{vis}} \times D}$. A learnable mask token $\mathbf{e}^{\text{mask}} \in \mathbb{R}^D$ is inserted at indices $\mathcal{M}_t$ to restore a length-$N_p$ sequence for decoding. Per frame, the encoder also outputs three prompt tokens $\mathbf{z}_t^{\text{id}}, \mathbf{z}_t^{\text{amb}}, \mathbf{z}_t^{\text{voc}} \in \mathbb{R}^D$. For audio, we use a pretrained $L$-layer speech encoder kept frozen during pretraining; let $\{\mathbf{H}^{(\ell)}\}_{\ell=1}^L$ denote its hidden-state sequences. To align with the $T$ video frames, we resample $\mathbf{a}$ so that the encoder emits $T$ tokens per utterance at every layer, yielding $\mathbf{A}^{(\ell)} = \{\mathbf{a}_t^{(\ell)}\}_{t=1}^T$.

### 3.2 Model Design

**Two-Bypass Face-Aware Masking.** Each frame is encoded twice with different masking *by-passes*. **Bypass 1 (Uniform)** applies a random 75% mask, yielding visible indices $\mathcal{V}_t^{(1)}$ and tokens $\mathbf{P}_t^{\text{vis},(1)} \in \mathbb{R}^{N_{\text{vis}} \times D}$. We decode from these visible tokens and the *identity* prompt $\mathbf{z}_t^{\text{id}}$, discarding the *vocal/ambient* prompts from this pass. **Bypass 2 (Face-Preserving)** also masks 75% of patches but retains facial regions with higher probability, producing $\mathcal{V}_t^{(2)}$. To preserve motion while suppressing

appearance leakage, we apply color/brightness/saturation perturbations and keep only the *vocal-motion* and *ambient-motion* prompts $\mathbf{z}_t^{\text{voc}}, \mathbf{z}_t^{\text{amb}}$ for decoding, discarding its visible patch tokens and identity prompt. In both passes the encoder consumes only visible patches; before decoding, we insert a learnable mask token at $\mathcal{M}_t^{(1)}$ (the complement of $\mathcal{V}_t^{(1)}$) to restore a length-$N_p$ sequence per frame, as in MAE.

**Visual Encoder and Prompt Tokens.** A ViT-style encoder applied to the visible patches outputs per-frame patch tokens (from Bypass 1) and three prompt tokens

$$\mathbf{Z}_t^{\text{prompt}} = \left[ \mathbf{z}_t^{\text{id}}, \ \mathbf{z}_t^{\text{amb}}, \ \mathbf{z}_t^{\text{voc}} \right] \in \mathbb{R}^{3 \times D},$$

where $\mathbf{z}_t^{\text{id}}$ is taken from Bypass 1 and $\mathbf{z}_t^{\text{voc}}, \mathbf{z}_t^{\text{amb}}$ from Bypass 2 as defined above.

**Identity-Consistent Prompt Shuffling.** Within a mini-batch, to promote motion-invariant identity embeddings, we randomly swap $\mathbf{z}^{\text{id}}$ among frames that share the same subject identity but exhibit different facial motions, weakening incidental coupling between identity prompts and short-term dynamics.

**Audio Features and Adapter.** A pretrained $L$-layer speech encoder (kept frozen) produces per-layer hidden sequences; we resample the input waveform so that each layer emits $T$ tokens (one per video frame), yielding aligned streams $\{\mathbf{a}_t^{(\ell)}\}_{t=1}^T$. To preserve both low-level acoustic detail and higher-level linguistic cues, we concatenate the aligned per-layer tokens along the feature dimension at each time step,

$$\tilde{\mathbf{a}}_t = \left[ \mathbf{a}_t^{(1)} \| \cdots \| \mathbf{a}_t^{(L)} \right],$$

and pass them through an audio adapter that projects to the shared width $D$, producing $\mathbf{A}_t \in \mathbb{R}^D$. This design mitigates the bias of top encoder layers toward semantic content induced by ASR pretraining while retaining waveform-proximal information from earlier layers.

**Decoder with Prompt Cross-Attention.** We use an MAE-style decoder with $N_{\text{dec}}$ Transformer blocks. Its input at each frame is the restored token sequence formed by encoded visible patches from Bypass 1 plus mask tokens (with positional embeddings). Each block first applies self-attention over this full sequence (visible patches + mask tokens), and then cross-attends to the prompt tokens—identity, ambient, and a conditioning token $\mathbf{c}_t$. We perform two decoding passes per frame that share decoder weights: (i) a video-driven pass with $\mathbf{c}_t = \mathbf{z}_t^{\text{voc}}$ producing $\hat{\mathbf{x}}_t^{\text{voc}}$, and (ii) an audio-driven pass with $\mathbf{c}_t = \mathbf{A}_t$ producing $\hat{\mathbf{x}}_t^{\text{aud}}$. Both passes reconstruct the same target frame $\mathbf{x}_t$, providing symmetric supervision that encourages the vocal-motion and audio tokens to encode consistent, alignable information. A linear head maps decoded tokens back to pixels.

**Prompt Token Factorization Analysis.** To examine whether the three prompts specialize into identity, vocal motion, and ambient motion, we conduct two analyses. First, we reconstruct masked portraits while sourcing $\mathbf{z}^{\text{id}}$, $\mathbf{z}^{\text{amb}}$, and $\mathbf{z}^{\text{voc}}$ from three frames with mismatched identity and expression (figure 2a) with the trained SyncLipMAE's MAE Decoder. The reconstructions follow the ambient source in eye blinks (via $\mathbf{z}^{\text{amb}}$) and the vocal source in mouth shape (via $\mathbf{z}^{\text{voc}}$), while $\mathbf{z}^{\text{id}}$ mainly controls static appearance. Second, we aggregate cross-attention weights from all decoder blocks and visualize per-prompt maps (figure 2b): attention from $\mathbf{z}^{\text{voc}}$ concentrates on the lower face(mainly on the mouth), $\mathbf{z}^{\text{amb}}$ spreads over the face with weak background response, and $\mathbf{z}^{\text{id}}$ focuses on background and non-mouth/eye facial regions. These patterns provide qualitative evidence that the three prompts form a factorized representation over identity, vocal motion, and ambient motion.

### 3.3 Pretraining Objectives

**Pixel-Space Reconstruction (MSE).** We minimize per-frame mean squared error for the two decoding passes (video-driven and audio-driven; see §3.2):

$$\mathcal{L}_{\text{pix}}^{\text{voc}} = \frac{1}{T} \sum_{t=1}^T \left\| \hat{\mathbf{x}}_t^{\text{voc}} - \mathbf{x}_t \right\|_2^2, \qquad \mathcal{L}_{\text{pix}}^{\text{aud}} = \frac{1}{T} \sum_{t=1}^T \left\| \hat{\mathbf{x}}_t^{\text{aud}} - \mathbf{x}_t \right\|_2^2.$$

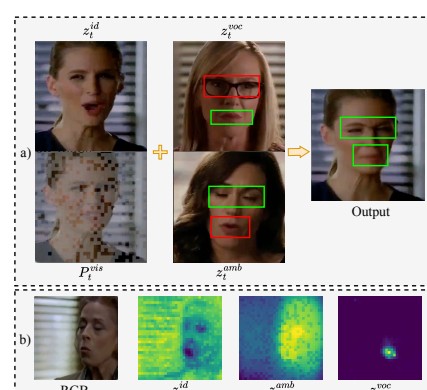 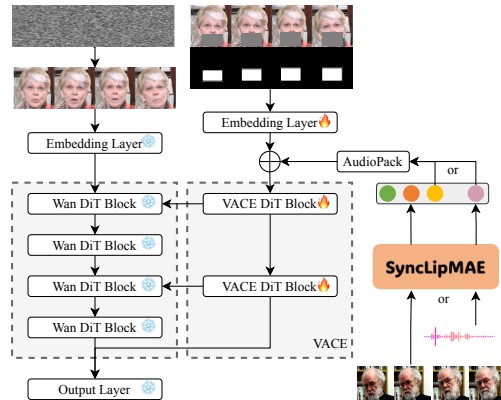

Figure 2: (a) SyncLipMAE MAE reconstructions under prompt-source swapping. (b) Cross-attention maps for $\mathbf{z}^{\text{id}}$, $\mathbf{z}^{\text{amb}}$, and $\mathbf{z}^{\text{voc}}$.

Figure 3: Our video dubbing architecture injects audio–visual synchronized feature to WanVACE DiT blocks.

**Audio–Vocal Contrastive Alignment.** Unlike prior audio–visual synchronisation systems that cast the task as in-/out-of-sync *binary* classification or as *pairwise* contrastive matching, we align per-frame audio tokens $\mathbf{A}_t$ with vocal-motion tokens $\mathbf{v}_s = \mathbf{z}_s^{\text{voc}}$ using a CLIP-style symmetric InfoNCE. Let $p_{t \to s} = \frac{\exp(\langle \mathbf{A}_t, \mathbf{v}_s \rangle / \tau)}{\sum_{u=1}^{T} \exp(\langle \mathbf{A}_t, \mathbf{v}_u \rangle / \tau)}$ be the softmax over vocal candidates for audio index $t$, and $P_t$ the positive set (aligned index and optional neighbors within $\pm k$, expanded by an audio self-similarity threshold). The loss is

$$\mathcal{L}_{\text{CL}} = -\frac{1}{2T} \sum_{t=1}^{T} \Big[ \log \sum_{s \in P_t} p_{t \to s} \ + \ \log \sum_{s \in P_t} p_{s \to t} \Big],$$

with temperature $\tau$. This preserves CLIP-style bidirectional alignment while allowing multiple positives per anchor.

**Cross-Covariance Decorrelation.** To reduce leakage across factor-specific embeddings, we penalize cross-covariance between centered tokens. For $\mathcal{P} = \{\text{id}, \text{amb}, \text{voc}\}$:

$$\mathcal{L}_{\text{cov}} = \frac{1}{D^2} \sum_{\{p,q\} \subset \mathcal{P}} \big\| \text{Cov}\big(\mathbf{z}^p, \mathbf{z}^q\big) \big\|_F^2,$$

where covariance is computed after centering along the batch/time axis.

**Total Objective.** The overall loss is

$$\mathcal{L} = \lambda_{\text{pix}} \big( \mathcal{L}_{\text{pix}}^{\text{voc}} + \mathcal{L}_{\text{pix}}^{\text{aud}} \big) + \lambda_{\text{CL}} \, \mathcal{L}_{\text{CL}} + \lambda_{\text{cov}} \, \mathcal{L}_{\text{cov}}.$$

## 4 DOWNSTREAM ADAPTATION

We apply SyncLipMAE to four tasks (see Tab 2 for the downstream heads and tokens used in each case): **(i) audio–visual stream synchronization** — we estimate frame-level lag and synchrony in the frozen shared space by maximizing cosine similarity between aligned audio tokens $\mathbf{A}_t$ and vocal-motion tokens $\mathbf{z}_t^{\text{voc}}$, without any additional head; **(ii) facial understanding** — we infer emotion and head/face actions by averaging the vocal-

Table 2: Used heads and prompt tokens.

| Task | Downstream head | Tokens |
|---|---|---|
| AV synchronisation | Similarity (no head) | $\mathbf{A}_t, \mathbf{z}_t^{\text{voc}}$ |
| Facial understanding | Linear classifier | $\mathbf{z}_t^{\text{voc}}, \mathbf{z}_t^{\text{amb}}$ |
| Lip reading (VSR) | Conformer+Transformer | $\mathbf{z}_t^{\text{voc}}$ |
| Video dubbing | WanVACE+AudioPack | $\mathbf{A}_t / \mathbf{z}_t^{\text{voc}}$ |

Table 1: Lip synchronisation on **Hallo3**. Higher is better for Acc and R-Precision; lower is better for Offset.

| Method | K = 1 | | K = 5 | | K = 15 | | R-precision (32) | | |
|---|---|---|---|---|---|---|---|---|---|
| | Acc ↑ (%) | Offset ↓ | Acc ↑ (%) | Offset ↓ | Acc ↑ (%) | Offset ↓ | Top1 ↑ | Top2 ↑ | Top3 ↑ |
| SyncNet -5 | 20.37 | 5.04 | 24.18 | 4.03 | 30.03 | 3.02 | 7.47 | 12.77 | 17.72 |
| VocaLiST -5 | 21.85 | 4.34 | 24.56 | 3.63 | 28.82 | 2.98 | 8.80 | 15.08 | 19.86 |
| StableSyncNet -16 | 27.95 | 2.78 | 28.60 | 2.72 | 31.66 | 2.72 | 14.40 | 25.65 | 34.52 |
| **SyncLipMAE -1 (ours)** | **52.53** | **2.66** | **68.41** | **1.73** | **82.27** | **0.93** | **40.18** | **61.48** | **73.49** |

and ambient-motion tokens per frame and feeding a single linear classifier; **(iii) visual speech recognition (VSR)** — we infer text from the vocal-motion tokens using a Conformer encoder and Transformer decoder trained with a CTC/attention hybrid loss; **(iv) visual dubbing** — we edit a source face video to match target speech or a reference motion by conditioning a WanVACE DiT backbone via an AudioPack adapter that injects either audio tokens $\mathbf{A}_t$ or reference vocal-motion tokens $\mathbf{z}_t^{\mathrm{voc}}$, supporting audio- or video-driven control within a single model.

## 4.1 AUDIO–VISUAL SYNCHRONISATION

We estimate audio–video lag directly in the pretrained shared space without any extra head or training. Given per-frame audio tokens $\mathbf{A}_t$ and vocal-motion tokens $\mathbf{z}_s^{\mathrm{voc}}$, compute frame-wise cosine similarity

$$s(t,s) = \langle \mathbf{A}_t, \mathbf{z}_s^{\mathrm{voc}} \rangle,$$

and aggregate along temporal offsets

$$S(\Delta) = \frac{1}{T-|\Delta|} \sum_{t=1}^{T-|\Delta|} s(t, t+\Delta), \qquad \hat{\Delta} = \arg \max_{\Delta \in [-\Delta_{\max}, \Delta_{\max}]} S(\Delta).$$

The in-sync score is $S(0)$ and the estimated lag is $\hat{\Delta}$. All computations use the frozen tokens from pretraining; no correspondence MLP or additional supervision is introduced.

## 4.2 FACIAL UNDERSTANDING: EMOTION AND ACTION

We keep SyncLipMAE frozen and attach a minimal linear head. For each frame $t$, we form a motion descriptor by averaging the two motion prompts,

$$\mathbf{u}_t = \tfrac{1}{2}\big(\mathbf{z}_t^{\mathrm{voc}} + \mathbf{z}_t^{\mathrm{amb}}\big),$$

feed $\mathbf{u}_t$ into a single linear layer to obtain frame-level logits, and then average the logits over sampled frames before applying a softmax to obtain video-level predictions. The identity prompt is not used in this head, so the classifier operates purely on motion factors for emotion/action classification.

## 4.3 LIP READING (VSR)

We adopt the visual speech recognition head of AutoAVSR (Ma et al., 2023) for lip reading and feed it with SyncLipMAE's vocal-motion tokens. Concretely, let $\mathbf{V} = \{\mathbf{z}_t^{\mathrm{voc}}\}_{t=1}^{T'}$ be the input sequence to a Conformer encoder (as in AutoAVSR), which produces hidden states $\mathbf{H} = \{\mathbf{h}_t\}_{t=1}^{T'}$. A Transformer decoder then autoregressively predicts subword units with cross-attention over $\mathbf{H}$, following the standard encoder–decoder formulation.

*Losses.* Let the target token sequence be $Y = \{y_u\}_{u=1}^{U}$ over vocabulary $\Sigma$ (blank $\varnothing$ for CTC). The CTC branch defines

$$P_{\mathrm{CTC}}(Y \mid \mathbf{H}) = \sum_{\pi \in \mathcal{B}^{-1}(Y)} \prod_{t=1}^{T'} p_{\mathrm{ctc}}(\pi_t \mid \mathbf{h}_t), \qquad \mathcal{L}_{\mathrm{CTC}} = -\log P_{\mathrm{CTC}}(Y \mid \mathbf{H}),$$

where $\mathcal{B}$ collapses repeats and removes blanks. The decoder branch is trained with teacher forcing:

$$\mathcal{L}_{\mathrm{DEC}} = -\sum_{u=1}^{U} \log p_{\mathrm{dec}}(y_u \mid y_{<u}, \mathbf{H}).$$

Table 3: Lip synchronisation on **VFHQ**.

| Method | K = 1 | | K = 5 | | K = 15 | | R-precision (32) | | |
|---|---|---|---|---|---|---|---|---|---|
| | Acc ↑ (%) | Offset ↓ | Acc ↑ (%) | Offset ↓ | Acc ↑ (%) | Offset ↓ | Top1 ↑ | Top2 ↑ | Top3 ↑ |
| SyncNet -5 | 22.16 | 5.14 | 26.47 | 4.34 | 31.09 | 3.29 | 9.40 | 16.91 | 23.09 |
| VocaLiST -5 | 21.76 | 5.14 | 24.34 | 4.57 | 29.23 | 3.62 | 10.18 | 16.56 | 21.98 |
| StableSyncNet -16 | 32.40 | **3.17** | 33.06 | 3.07 | 35.43 | 2.90 | 23.54 | 37.65 | 46.77 |
| **SyncLipMAE -1 (ours)** | **38.58** | 3.80 | **48.95** | **3.06** | **61.78** | **2.16** | **35.52** | **52.98** | **63.37** |

The total objective follows the standard hybrid CTC/attention form used in AutoAVSR,

$$\mathcal{L}_{\text{VSR}} = \lambda_{\text{CTC}}\,\mathcal{L}_{\text{CTC}} + (1 - \lambda_{\text{CTC}})\,\mathcal{L}_{\text{DEC}},$$

with a tunable weight $\lambda_{\text{CTC}} \in (0, 1)$.

### 4.4 VIDEO DUBBING

We cast dubbing as masked video inpainting on face crops within a VACE-style unified interface: task inputs (reference frames and a spatiotemporal mouth mask) are packed into a Video Condition Unit and processed by a Diffusion-Transformer backbone with Concept Decoupling and Context Adapters (Jiang et al., 2025). We instantiate this pipeline on WanVACE, keeping its architecture, losses, and training schedule unchanged; WanVACE natively accepts 1–3 reference images as a comma-separated list, which we use for identity control.

*Control injection.* We adopt an *AudioPack* placed after the VACE patch-embedding stage and before the first DiT VACE block, inspired by audio-conditioned injection in avatar generation (Gan et al., 2025). Per-frame control tokens—either audio tokens $\mathbf{A}_t$ from SyncLipMAE or reference vocal-motion tokens $\mathbf{z}_t^{\text{voc}}$—are temporally encoded, projected to the DiT width, and added to the patch-embedded video token stream. No other WanVACE components are modified; switching $\mathbf{A}_t$ vs. $\mathbf{z}_t^{\text{voc}}$ toggles audio-driven vs. video-driven control within the same model.

*Long-form continuity via dual references.* To support arbitrarily long, coherent dubbing, we provide two references: (i) an identity image (stable appearance prior) and (ii) the last frame of the previous segment (temporal bridge). During training we randomly drop the continuity reference with 10% probability so the model can also initialise the first segment without a history frame.

*Masking strategy.* We detect the facial region (chin to hairline) and mask its lower half as the editable area, expanding the mask slightly below the chin to accommodate jaw excursions. This follows common lower-face inpainting setups in Wav2Lip-style and diffusion-based lip-sync systems.

## 5 EXPERIMENTS

**Implementation details.** *Backbones.* The visual encoder follows the Sapiens-0.3B (Khirodkar et al., 2024) configuration and is initialized from its pretrained weights; the decoder is a ViTMAE-B with a cross-attention sublayer in every block attending to the identity/ambient/vocal prompts. The audio branch uses a pretrained wav2vec 2.0 (Baevski et al., 2020) base encoder, kept frozen during SyncLipMAE training. *SyncLipMAE training.* SyncLipMAE is trained on 128 GPUs with batch size 48 per GPU (global 128×48) for 400k steps (∼15 days), using loss weights $\lambda_{\text{pix}} = \lambda_{\text{cl}} = 1$ and $\lambda_{\text{cov}} = 0.1$, temporal_neighbors = 1 (3-frame positive window), and a self-similarity threshold of 0.9. At each step we randomly sample a batch of frames from full videos, allowing consecutive or non-consecutive frames. *Video dubbing fine-tuning.* For dubbing we initialize WanVACE-1.3B, freeze all Wan backbone blocks, and optimize only the VACE blocks (∼0.7B trainable parameters) for ∼30k steps (∼3 days) on 128 GPUs with batch size 1 per GPU. Each sample contains 17+1+1 frames: 17 target frames, one history frame (temporal continuity), and one identity frame (appearance prior).

**SyncLipMAE training data.** We pretrain SyncLipMAE on Hallo3 Cui et al. (2025), CelebV-HQ Zhu et al. (2022), CelebV-Text Yu et al. (2023), MEAD Wang et al. (2020), VFHQ Xie et al. (2022), HDTF Zhang et al. (2021), and ESS Livingstone & Russo (2018). Audio is converted to mono

Table 4: CelebV-HQ emotion and action classification. Higher is better (Accuracy/AUC).

| Method | Frames | Emotion | | Action | |
|---|---|---|---|---|---|
| | | Accuracy ↑ | AUC ↑ | Accuracy ↑ | AUC ↑ |
| VideoMAE | $1 \times 16$ | 68.63 | 60.32 | 93.97 | 84.31 |
| VideoMAE | $5 \times 16$ | 68.63 | 66.60 | 94.54 | 86.01 |
| MARLIN | $1 \times 16$ | 66.67 | 58.15 | 94.69 | 82.26 |
| MARLIN | $5 \times 16$ | 69.61 | 64.98 | 94.86 | 83.90 |
| **SyncLipMAE (ours)** | 16 | **76.47** | **73.94** | **95.06** | **86.45** |
| **SyncLipMAE (ours)** | 80 | **77.45** | **78.33** | **95.11** | **86.69** |

16 kHz and videos are resampled to 25 fps. For VFHQ and MEAD we follow the official evaluation protocols, using the released test partitions for evaluation and the remaining clips for pretraining, and additionally sampling 100 training clips for validation. For Hallo3, CelebV-HQ, CelebV-Text, HDTF, and RAVDESS, which lack standardized splits in our setting, we randomly sample 100 clips for validation and 100 clips for testing, and use the rest for pretraining. All videos are decoded, uniformly resized, and center-cropped to $512 \times 512$, ensuring that the full face remains.

**Audio–visual stream synchronization.** We evaluate alignment under two complementary protocols that require no additional training: (i) *temporal lag detection* by sliding the offset between the audio token sequence $A = \{a_t\}_{t=1}^{T}$ and the visual token sequence $V = \{v_t\}_{t=1}^{T}$, and (ii) *audio→video token matching* reported as R-precision@$k$.

*Temporal lag detection.* For a candidate lag $\tau \in \mathcal{T}$ (positive: video lags audio), define the average distance

$$D(\tau) = \frac{1}{T_\tau} \sum_{t=1}^{T_\tau} d\big(a_t, v_{t+\tau}\big),$$

with $T_\tau$ chosen so indices are valid and $d(\cdot, \cdot)$ a feature distance (e.g., cosine or $\ell_2$). The estimated lag is

$$\hat{\tau} = \arg\min_{\tau \in \mathcal{T}} D(\tau).$$

Given ground-truth lag $\tau^\star$, we report

$$\text{Offset} = \tfrac{1}{N} \sum_{s=1}^{N} |\hat{\tau}^{(s)} - \tau^{(s)}| \quad \text{and} \quad \text{Acc}_{\pm K} = \tfrac{1}{N} \sum_{s=1}^{N} \mathbf{1}\big(|\hat{\tau}^{(s)} - \tau^{(s)}| \le K\big).$$

*Audio→Video token matching (R-precision@$k$).* We additionally report this retrieval-style metric because in segments with speaker silence or very small mouth motion the lag-based signal (Offset/Acc) can become weak or ambiguous, whereas a ranked retrieval test still probes cross-pair discriminability among distractors. Within a minibatch of size $B$, let $\{(a_i, v_i)\}_{i=1}^{B}$ be audio/visual token pairs and let $\mathcal{P}_i$ be the candidate pool consisting of the true $v_i$ plus non-matching video tokens (pool size fixed by the batch or a pre-defined sampler). Rank $\mathcal{P}_i$ by distance to $a_i$ and denote the rank of $v_i$ by $\text{rank}_i(v_i)$ (1 is best). The retrieval metric is

$$\text{R-precision@}k = \frac{1}{B} \sum_{i=1}^{B} \mathbf{1}\big(\text{rank}_i(v_i) \le k\big),$$

reported for small $k$ (e.g., $k = 1, 2, 3$) and fixed pool size (e.g., 32).

*Results.* Tab 1 and Tab 3 summarise results on Hallo3 and VFHQ. Across both datasets, SyncLip-MAE consistently improves temporal alignment (higher $\text{Acc}_{\pm K}$, lower *Offset*) and cross-modal retrieval (higher R-precision@$k$). Notably, these gains are achieved with single-frame visual conditioning ($n=1$), whereas baselines rely on wider temporal windows ($n=5$ or 16), indicating stronger per-frame audio–visual correspondence and reduced dependence on temporal aggregation.

Table 5: Video dubbing on **VFHQ** and **Hallo3**. **A-** denotes audio-driven control; **V-** denotes video-driven control. Higher is better for $\text{Sync}_{\text{conf}}$; lower is better for FID/FVD.

| Method | VFHQ | | | Hallo3 | | |
|---|---|---|---|---|---|---|
| | $\text{Sync}_{\text{conf}}\uparrow$ | FID$\downarrow$ | FVD$\downarrow$ | $\text{Sync}_{\text{conf}}\uparrow$ | FID$\downarrow$ | FVD$\downarrow$ |
| MuseTalk Zhang et al. (2024b) | 0.6820 | 24.44 | 175.16 | 0.6904 | 13.80 | 86.46 |
| LatentSync-1.5 Li et al. (2024) | 0.7681 | 17.92 | 131.04 | 0.7725 | 10.59 | 78.89 |
| **A-SyncLipMAE (ours)** | **0.7830** | **16.46** | **100.44** | **0.8027** | **10.04** | **74.01** |
| **V-SyncLipMAE (ours)** | – | 17.22 | 109.91 | – | 10.82 | **77.57** |

**Facial understanding (emotion & actions).** We evaluate on **CelebV-HQ** (Zhu et al., 2022) using the *emotion* and *action* attribute sets, reporting video-level Accuracy and AUC. All classifiers are trained on the CelebV-HQ training split and evaluated on the test split. Emotion is an 8-way single-label task, whereas action is multi-label over 35 categories (one-vs-rest at the video level). Following common practice (Tong et al., 2022; Korbar et al., 2019), clip-based baselines (e.g., VideoMAE/MARLIN) use an $a \times b$ protocol—uniformly sampling $a$ clips of $b$ frames and averaging clip predictions; frame-averaging models sample $n$ frames and average per-frame predictions. All methods attach a single linear classifier to frozen features: given sampled clips or frames $\{\mathbf{x}_i\}_{i=1}^m$, encoder $f$, and head $W$,

$$\bar{\ell}(V) = \frac{1}{m} \sum_{i=1}^m W f(\mathbf{x}_i),$$

with video-level predictions and Accuracy/AUC computed from $\bar{\ell}(V)$.

*Results.* Tab 4 summarizes CelebV-HQ emotion and action. On emotion, SyncLipMAE clearly surpasses the best baseline with 16 frames and improves further at 80 frames, indicating benefits from longer temporal context. On action—where baselines are already strong—SyncLipMAE provides consistent but modest gains, with a slight boost when using 80 frames. Overall, SyncLipMAE delivers substantial improvements on emotion and steady gains on action.

Table 6: Visual speech recognition on **VFHQ**, **Hallo3**, and **HDTF**. Lower is better (WER$\downarrow$).

| Method | WER$\downarrow$ | | |
|---|---|---|---|
| | **VFHQ** | **Hallo3** | **HDTF** |
| AV-HuBERT Shi et al. (2022) | 14.29 | 16.57 | 15.28 |
| Auto-AVSR Ma et al. (2023) | 8.13 | 6.23 | 5.91 |
| **SyncLipMAE** | **7.37** | **5.99** | **5.78** |

**Visual Speech Recognition (VSR)** We evaluate SyncLipMAE on HDTF, VFHQ, and Hallo3 for visual-only VSR. A single VSR head is trained jointly on the training splits of all three datasets, using speech transcripts obtained by running the Whisper Turbo model (Radford et al., 2023) on the original audio, and evaluated on the corresponding test splits after filtering out non-English utterances. Following standard practice, we report Word Error Rate (WER), computed from word-level substitutions ($S$), deletions ($D$), and insertions ($I$) against a reference of length $N$:

$$\text{WER} = \frac{S + D + I}{N}.$$

*Results.* Tab 6 reports WER on VFHQ, Hallo3, and HDTF. For AV-HuBERT, we use the official *Large + Self-Training* checkpoint pretrained on LRS3 + VoxCeleb2 (En) and finetuned on LRS3-433h. The visual-only head built on SyncLipMAE 's tokens achieves the lowest WER on all three sets: 7.37 on VFHQ, 5.99 on Hallo3, and 5.78 on HDTF, improving over Auto-AVSR by absolute 0.76, 0.24, and 0.13, respectively, and outperforming AV-HuBERT on each set. These results indicate that SyncLipMAE 's pretraining yields visual motion features that transfer effectively to sentence-level lip reading under the standard hybrid CTC/attention recipe.

**Video dubbing.** We train a WanVACE-based *video dubbing* model on the same mixture of talking-face corpora used for SyncLipMAE pretraining and evaluate it on the HDTF, Hallo3, and VFHQ test splits, re-synchronising the mouth region of a source video to a target speech while preserving identity, pose, and expressions. The model follows a VACE-style unified interface and supports both audio- and video-driven conditioning via a lightweight vocal-cue injection (details in section 4.4).

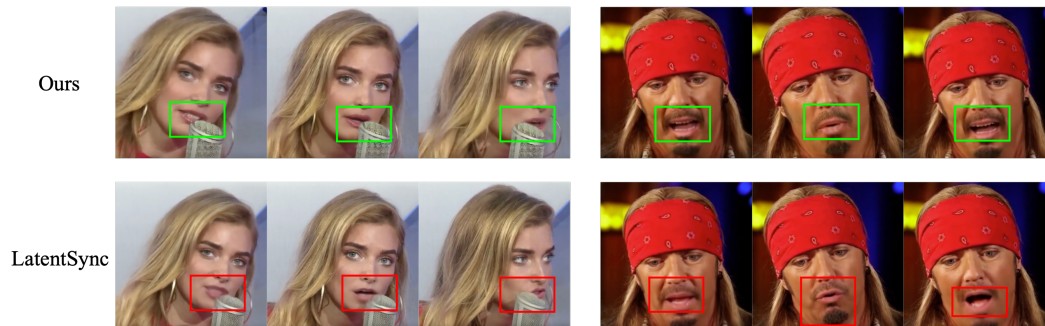

Figure 4: Qualitative visual comparison: LatentSync vs. our approach.

We report the metrics in Tab 5. $\text{Sync}_{\text{conf}}$. Identical to LSE-C, it is computed with the *stable SyncNet* evaluator from LatentSync (Li et al., 2024), instead of the vanilla SyncNet whose scores we found less reliable (Chung & Zisserman, 2016). Following SyncNet, for each sliding window we form an audio–video distance curve over temporal offsets $\Delta \in [-L, L]$:

$$d_t(\Delta) = \big\| \phi_a\big(a_{t:t+w-1}\big) - \phi_v\big(v_{t+\Delta:\,t+\Delta+w-1}\big) \big\|_2,$$

and define the per-window confidence as the median–minimum gap,

$$\text{Conf}_t = \text{median}_\Delta\, d_t(\Delta) - \min_\Delta\, d_t(\Delta), \qquad \text{Sync}_{\text{conf}} = \frac{1}{T} \sum_{t=1}^{T} \text{Conf}_t,$$

where larger values indicate a clearer minimum at the correct offset (stronger A/V synchrony). **FID** measures per-frame perceptual quality via the Fréchet distance between Inception features of real and generated frames (lower is better) (Heusel et al., 2017), and **FVD** extends this to spatiotemporal video features to jointly capture visual quality and temporal coherence (lower is better) (Unterthiner et al., 2018).

**Results.**    Qualitatively (Fig. 4), SyncLipMAE preserves identity and fine lip details (e.g., teeth, beard) and is more robust to occlusions than LatentSync/MuseTalk. Quantitatively (Tab. 5), consistent with these observations, the audio-driven variant (**A-SyncLipMAE**) attains the best overall scores on synchrony ($\text{Sync}_{\text{conf}}$) and perceptual quality (FID/FVD) on VFHQ and Hallo3. The video-driven variant (**V-SyncLipMAE**) closely matches the audio-driven performance and surpasses prior methods, indicating that the shared token space indeed aligns modalities and enables no-difference audio- or video-driven control within a single model.

**Ablation Studies.**    We perform ablations on the key components of SyncLipMAE, examining: (i) whether to enable Two-Bypass Face-Aware Masking; (ii) the strategy for adapting audio features; (iii) the inclusion of cross-attention in the decoder; and (iv) the composition of the prompt tokens. Detailed results are provided in the Appendix.

## 6    CONCLUSION

We introduced SyncLipMAE, a self-supervised framework for talking-face video that couples masked visual reconstruction with contrastive audio–visual alignment and explicitly factorizes each frame into three prompt tokens—**identity**, **vocal motion**, and **ambient motion**. This design yields synchronization-aware, disentangled features that transfer directly to diverse applications. Across four tasks—audio–visual stream synchronization, facial understanding, visual speech recognition, and unified visual dubbing—SyncLipMAE achieves state-of-the-art performance, validating the effectiveness of this pretraining strategy.

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

Table 7: Statistics of the talking-face pretraining corpus and the downstream tasks that each dataset is used for. Durations are reported in hours.

| Dataset | #Clips | Duration (h) | Pretrain (AV sync) | Emotion / action cls. | VSR | Video dubbing |
|---|---|---|---|---|---|---|
| HDTF | 5,433 | 14.51 | ✓ | ✕ | ✓ | ✓ |
| MEAD | 203,013 | 243.78 | ✓ | ✕ | ✕ | ✓ |
| RAVDNESS | 1,440 | 1.51 | ✓ | ✕ | ✕ | ✓ |
| CelebV-HQ | 35,662 | 68.07 | ✓ | ✓ | ✕ | ✓ |
| CelebV-Text | 67,005 | 261.23 | ✓ | ✕ | ✕ | ✓ |
| Hallo3 | 100,229 | 132.03 | ✓ | ✕ | ✓ | ✓ |
| VFHQ | 15,174 | 36.94 | ✓ | ✕ | ✓ | ✓ |
| Total | 427,956 | 758.07 | – | – | – | – |

## A  DATASET STATISTICS

Our pretraining corpus aggregates 427,956 talking-face clips spanning approximately 758 hours of video from seven public datasets: HDTF, MEAD, RAVDESS, CelebV-HQ, CelebV-Text, Hallo3, and VFHQ. As summarized in Tab 7, HDTF, MEAD, RAVDESS, CelebV-HQ, CelebV-Text, Hallo3, and VFHQ contribute 14.51, 243.78, 1.51, 68.07, 261.23, 132.03, and 36.94 hours respectively. All seven datasets are used for our self-supervised pretraining (AV sync), while different subsets are activated for downstream evaluation: CelebV-HQ is the only source for emotion/action classification, HDTF/Hallo3/VFHQ support VSR benchmarks, and all datasets are used for video dubbing.

The datasets differ markedly in acquisition conditions and semantic coverage. HDTF mainly contains high-definition frontal talking-head videos of news-like broadcasts, offering clean lip motion and relatively simple backgrounds that are well-suited for lip synchronization and VSR. MEAD and RAVDESS are studio-recorded emotional corpora where multiple actors read fixed sentences under controlled lighting and camera setups with discrete emotion labels; MEAD further provides multi-view recordings from several camera angles, while RAVDESS focuses on carefully validated emotional speech and song. CelebV-HQ and CelebV-Text are large-scale, in-the-wild celebrity video datasets with rich annotations (attributes or paired texts) and diverse appearance, head poses, and actions; however, the subjects are not always strictly "talking"—clips may include singing, non-verbal expressions, or other activities—so we primarily rely on them for pretraining and video dubbing rather than VSR supervision. Hallo3 provides high-quality, highly dynamic portrait videos with varied camera viewpoints and rich real-world scenes (e.g., interviews, TV dramas, conversational clips), where audio can contain background sounds in addition to speech, making it particularly suitable for stress-testing robustness in VSR and dubbing. VFHQ is a high-fidelity video face dataset originally curated for video face super-resolution and consists mainly of diverse interview-style talking heads, including singing and non-English speech, which we repurpose as another strong source of high-quality portrait footage for pretraining, VSR, and dubbing.

Overall, combining these sources yields a corpus that balances scale and diversity across identities, poses, emotions, recording environments, and linguistic content. This mix is crucial for learning synchronization-aware facial dynamics that transfer reliably from controlled studio-style datasets (MEAD/RAVDESS) to in-the-wild, multi-scene talking-head footage (CelebV-HQ, CelebV-Text, Hallo3, VFHQ), and for supporting a broad suite of downstream tasks spanning AV synchronization, emotion/action classification, VSR, and video dubbing.

## B  ABLATION STUDIES

**Two-Bypass Face-Aware Masking.**  We evaluate four variants on **Hallo3** using the same A/V synchronisation protocol as the main results ($\text{Acc}_{\pm K}$, Offset, R-precision; see Tab 8). *A1 (Uniform-only)* severely underperforms: heavy uniform masking hides the lip region, the contrastive loss scarcely decreases, and all sync metrics degrade markedly. *A2 (Face-aware-only)* trains the contrastive objective but yields weaker reconstruction guidance, producing normal yet inferior sync scores. *A3 (Two-bypass w/o photometric)* restores learning stability but, without motion-view photometric jitter, motion tokens leak appearance; alignment improves over A2 but remains below the best. *A4 (Two-bypass + photometric, ours)* is strongest across all metrics, confirming that a uniform view for

Table 8: Ablations on **Hallo3** (audio–visual stream synchronization). Higher is better for Acc and R-Precision; lower is better for Offset.

| Variant | K = 1 | | K = 5 | | K = 15 | | R-precision (32) | | |
|---|---|---|---|---|---|---|---|---|---|
| | Acc ↑ (%) | Offset ↓ | Acc ↑ (%) | Offset ↓ | Acc ↑ (%) | Offset ↓ | Top1 ↑ | Top2 ↑ | Top3 ↑ |
| *(A) Two-Bypass Face-Aware Masking* | | | | | | | | | |
| A1  Uniform-only (single view) | 9.34 | 7.18 | 12.68 | 6.22 | 18.57 | 5.36 | 2.83 | 5.14 | 7.92 |
| A2  Face-aware-only (single view) | 45.63 | 3.45 | 61.07 | 2.37 | 75.88 | 1.27 | 33.45 | 52.81 | 66.19 |
| A3  Two-bypass w/o photometric | 49.62 | 3.02 | 65.91 | 2.03 | 79.87 | 1.05 | 37.54 | 57.89 | 70.86 |
| *(B) Audio Feature Adaptation* | | | | | | | | | |
| B1  Last-layer only | 42.68 | 3.62 | 58.34 | 2.51 | 73.45 | 1.36 | 30.92 | 49.87 | 63.11 |
| *(C) Decoder Cross-Attention* | | | | | | | | | |
| C1  No cross-attention | 48.27 | 3.10 | 64.31 | 2.09 | 78.44 | 1.09 | 36.21 | 56.48 | 69.32 |
| *(D) Prompt Tokens* | | | | | | | | | |
| D1  $z^{\mathrm{voc}}$ | 47.12 | 3.18 | 63.27 | 2.16 | 77.95 | 1.12 | 35.64 | 55.73 | 68.41 |
| D2  $z^{\mathrm{id}}, z^{\mathrm{voc}}$ | 50.03 | 2.90 | 66.47 | 1.96 | 80.56 | 1.00 | 38.22 | 59.03 | 71.92 |
| D3  $z^{\mathrm{amb}}, z^{\mathrm{voc}}$ | 51.41 | 2.74 | 67.39 | 1.82 | 81.33 | 0.97 | 39.26 | 60.37 | 72.81 |
| **Ours;** $z^{\mathrm{id}}, z^{\mathrm{voc}}, z^{\mathrm{amb}}$ | **52.53** | **2.66** | **68.41** | **1.73** | **82.27** | **0.93** | **40.18** | **61.48** | **73.49** |

Table 9: Lip synchronisation on **HDTF**.

| Method | K = 1 | | K = 5 | | K = 15 | | R-precision (32) | | |
|---|---|---|---|---|---|---|---|---|---|
| | Acc ↑ (%) | Offset ↓ | Acc ↑ (%) | Offset ↓ | Acc ↑ (%) | Offset ↓ | Top1 ↑ | Top2 ↑ | Top3 ↑ |
| SyncNet | 37.91 | 3.91 | 46.92 | 2.70 | 55.10 | 1.66 | 20.81 | 32.92 | 41.83 |
| VocaLiST -LRS2 | 40.61 | 3.57 | 46.99 | 2.77 | 54.94 | 1.80 | 24.08 | 36.04 | 43.75 |
| StableSyncNet | **54.24** | **1.64** | 55.51 | **1.56** | 57.04 | 1.46 | **46.77** | 62.26 | 71.11 |
| **SyncLipMAE (ours)** | 44.98 | 3.04 | **57.70** | 2.09 | **69.67** | **1.29** | 43.70 | **63.52** | **74.41** |

reconstruction context plus a face-preserving view—with photometric perturbations—for prompt tokens best supports token-level A/V alignment.

**Audio Feature Adaptation.**    We compare *B1 (Last-layer only)* versus *B2 (Concat→Adapter, ours)*. B1 is consistently worse than A2 and B2: using only the wav2vec 2.0 final layer emphasizes semantic content and attenuates fine-grained timing, leading to lower $\mathrm{Acc}_{\pm K}$, higher Offset, and reduced R-precision. B2 concatenates all hidden layers and adapts them to the visual width, yielding the highest synchronization scores among the audio variants.

**Decoder Cross-Attention.**    We ablate *C1 (No cross-attention)* against *C2 (CA to id+amb+$c_t$, ours)*. In C1, when prompt tokens are simply concatenated with patch tokens before self-attention, the decoder tends to ignore them, slightly underperforming A3. C2 explicitly cross-attends to identity, ambient, and the conditioning token $c_t \in \{\mathbf{z}_t^{\mathrm{voc}}, \mathbf{A}_t\}$, which improves A/V alignment and yields the best overall sync metrics.

**The Composition of the Prompt Tokens**    As introduced in section 3.1, SyncLipMAE encodes each talking-face frame into a three-token representation $(\mathbf{z}^{\mathrm{id}}, \mathbf{z}^{\mathrm{amb}}, \mathbf{z}^{\mathrm{voc}})$. In this experiment, we examine the necessity and effectiveness of this design by ablating the prompt-token composition: we remove $\mathbf{z}^{\mathrm{id}}$, $\mathbf{z}^{\mathrm{amb}}$, or both, and then evaluate how these variants affect the quality of the extracted $\mathbf{z}^{\mathrm{voc}}$ on the AV-Sync task. As shown in Experiment D, dropping any of the tokens consistently degrades the

Table 10: Audio–visual stream synchronization on **CelebV-HQ**. Higher is better for Acc/R-precision; lower is better for Offset.

| Smooth window | K=1 | | K=5 | | K=15 | | R-precision (32) | | |
|---|---|---|---|---|---|---|---|---|---|
| Method | Acc ↑ (%) | Offset ↓ | Acc ↑ (%) | Offset ↓ | Acc ↑ (%) | Offset ↓ | Top1 ↑ | Top2 ↑ | Top3 ↑ |
| SyncNet | 20.26 | 6.14 | 24.50 | 5.67 | 32.07 | 4.99 | 8.08 | 14.90 | 20.91 |
| VocaLiST – LRS2 | 17.42 | 6.60 | 20.46 | 6.38 | 25.96 | 5.89 | 5.98 | 11.59 | 16.45 |
| StableSyncNet | 32.95 | 4.66 | 34.35 | 4.62 | 38.04 | 4.44 | 21.26 | 33.20 | 40.69 |
| **Ours** | 32.63 | 4.83 | **40.96** | **4.41** | **52.91** | **3.56** | **28.47** | **44.19** | **54.32** |

Table 11: Audio–visual stream synchronization on **CelebV-Text**. Higher is better for Acc/R-precision; lower is better for Offset.

| Smooth window | K=1 | | K=5 | | K=15 | | R-precision (32) | | |
|---|---|---|---|---|---|---|---|---|---|
| Method | Acc ↑ (%) | Offset ↓ | Acc ↑ (%) | Offset ↓ | Acc ↑ (%) | Offset ↓ | Top1 ↑ | Top2 ↑ | Top3 ↑ |
| SyncNet | 18.80 | 6.10 | 22.73 | 5.40 | 28.85 | 4.56 | 8.89 | 15.03 | 20.54 |
| VocaLiST – LRS2 | 22.05 | 5.87 | 25.96 | 5.39 | 32.27 | 4.64 | 11.08 | 17.66 | 23.04 |
| StableSyncNet | 34.42 | 3.93 | 35.51 | 3.88 | 37.58 | 3.76 | 27.02 | 37.25 | 43.35 |
| **Ours** | **34.95** | 5.10 | **46.46** | 4.44 | **60.60** | **3.51** | **25.80** | **39.02** | **47.99** |

Table 12: Audio–visual stream synchronization on **MEAD**. Higher is better for Acc/R-precision; lower is better for Offset.

| Smooth window | K=1 | | K=5 | | K=15 | | R-precision (32) | | |
|---|---|---|---|---|---|---|---|---|---|
| Method | Acc ↑ (%) | Offset ↓ | Acc ↑ (%) | Offset ↓ | Acc ↑ (%) | Offset ↓ | Top1 ↑ | Top2 ↑ | Top3 ↑ |
| SyncNet | 39.39 | 4.01 | 50.03 | 2.97 | 61.18 | 1.85 | 20.74 | 32.29 | 40.01 |
| VocaLiST – LRS2 | 35.02 | 4.59 | 43.81 | 3.64 | 55.60 | 2.50 | 15.69 | 26.03 | 33.61 |
| StableSyncNet | **57.51** | **2.05** | **60.36** | **1.85** | 64.25 | 1.57 | 37.20 | 54.19 | 62.83 |
| **Ours** | 45.90 | 2.77 | 56.20 | 2.04 | **69.76** | **1.30** | **38.63** | **60.94** | **74.36** |

ability of $\mathbf{z}^{\text{voc}}$ to match the audio tokens, which we attribute to the orthogonality loss in our objective that explicitly encourages the three factors to focus on complementary aspects of the talking-face frame.

## C MORE RESULTS

**Audio–visual stream synchronization.** We additionally evaluate SyncLipMAE on **CelebV-HQ**, **CelebV-Text**, **RAVDESS**, and **MEAD**, and report these results in the Appendix. We do so because these corpora contain substantial non-speech or highly repeated content that can confound lag metrics and retrieval: CelebV-HQ and CelebV-Text are broad, face-centric video sets with attributes beyond active speech (appearance, actions, emotions, diverse in-the-wild clips), so many segments are not strictly speaking-focused; RAVDESS and MEAD are acted emotion datasets captured in controlled settings, with RAVDESS using only two fixed sentences across many takes (plus sung versions), leading to heavy lexical repetition.

**Zero-shot Comparison** Given that prior state-of-the-art methods and our approach differ in both training procedure and training data sources, it is important to compare them under a truly zero-shot setting on datasets that none of the methods have seen during training. For AV synchronization, we therefore collected 100 public speaking videos from the internet and evaluated all methods in a zero-shot manner on this set. For VSR, neither our model nor the baselines are trained on CelebV-HQ

Table 15: Zero-shot Visual speech recognition on **CelebV-HQ**, **CelebV-Text**, and **RAVDESS**. Lower is better (WER↓).

| Method | WER↓ | | |
|---|---|---|---|
| | CelebV-HQ | CelebV-Text | RAVDESS |
| AV-HuBERT Shi et al. (2022) | 15.17 | 16.73 | 3.09 |
| Auto-AVSR Ma et al. (2023) | 10.39 | 12.80 | 2.45 |
| **SyncLipMAE** | **10.05** | 12.19 | **2.42** |

Table 13: Audio–visual stream synchronization on **RAVDESS**. Higher is better for Acc/R-precision; lower is better for Offset.

| Smooth window | K=1 | | K=5 | | K=15 | | R-precision (32) | | |
|---|---|---|---|---|---|---|---|---|---|
| Method | Acc ↑ (%) | Offset ↓ | Acc ↑ (%) | Offset ↓ | Acc ↑ (%) | Offset ↓ | Top1 ↑ | Top2 ↑ | Top3 ↑ |
| SyncNet | 20.54 | 9.38 | 26.76 | 8.79 | 38.52 | 7.54 | 14.47 | 22.69 | 28.48 |
| VocaLiST – LRS2 | 22.12 | 8.93 | 27.97 | 8.33 | 39.96 | 7.15 | 14.94 | 23.46 | 29.21 |
| StableSyncNet | 42.69 | 6.91 | 46.06 | 6.62 | 53.79 | 5.97 | 34.51 | 46.21 | 51.28 |
| **Ours** | **47.94** | **3.09** | **61.03** | **2.37** | **79.03** | **1.23** | **39.97** | **62.90** | **76.26** |

Table 14: Lip synchronisation on **out-of-distribution data**. Higher is better for Acc and R-Precision; lower is better for Offset.

| Method | K = 1 | | K = 5 | | K = 15 | | R-precision (32) | | |
|---|---|---|---|---|---|---|---|---|---|
| | Acc ↑ (%) | Offset ↓ | Acc ↑ (%) | Offset ↓ | Acc ↑ (%) | Offset ↓ | Top1 ↑ | Top2 ↑ | Top3 ↑ |
| SyncNet -5 | 25.98 | 5.19 | 33.38 | 4.15 | 41.34 | 3.08 | 13.31 | 21.21 | 27.99 |
| VocaLiST -5 | 25.77 | 5.10 | 29.82 | 4.51 | 35.46 | 3.73 | 13.28 | 21.44 | 27.51 |
| StableSyncNet -16 | 36.20 | **3.19** | 37.53 | 3.10 | 40.49 | 2.89 | 27.37 | 38.57 | 46.50 |
| **SyncLipMAE -1 (ours)** | **40.30** | 3.78 | **50.71** | **3.02** | **61.67** | **2.25** | **33.66** | **51.12** | **61.33** |

or CelebV-Text, so we additionally constructed a zero-shot test set by selecting 50 clips with normal talking behaviour from these datasets. As shown in Tab 14 and Tab 15, SyncLipMAE also outperforms previous state-of-the-art methods on datasets it has never been trained on.

*Experimental analysis.* We observe that on the AV-Sync task, SyncLipMAE exhibits surprisingly strong performance. Despite operating on single frames only, SyncLipMAE shows consistent gains in both temporal alignment (higher $\text{Acc}_{\pm K}$, lower Offset) and cross-modal retrieval (higher R-precision@K). Thanks to being trained on talking-face datasets that cover a wide range of scenarios, this advantage is particularly pronounced in generic in-the-wild settings and in the presence of environmental noise (Hallo3, VFHQ, zero-shot evaluation). On some datasets (HDTF, MEAD), however, the purely single-frame design yields noisier behavior when we aggregate frame-wise predictions over a short temporal window, leading to slightly weaker temporal alignment than the multi-frame counterpart at $K=1$ and $K=5$. As $K$ increases, the advantages of SyncLipMAE become increasingly pronounced. For VSR, the well-aligned audio–visual representations learned by SyncLipMAE make it easier for the model to infer the underlying speech content from the observed visual cues.

# D LIMITATION

While SyncLipMAE achieves strong results across four disparate tasks, several limitations remain. (1) Model size and deployability. Compared with classic synchronisation backbones such as SyncNet—which is lightweight enough to be used directly as an evaluation head or even as a discriminator/loss in generation systems like Wav2Lip—SyncLipMAE is substantially heavier, making "drop-in loss" usage during training less practical and raising compute and memory costs for deployment.

(2) Scope of applicability. Although SyncLipMAE aligns audio and visual tokens well for lip-sync, expression/action understanding, and VSR, it is not a universal solution for all audio–face problems. For example, when we followed a speech-to-3DMM pipeline (in the spirit of audio-driven 3D talking-head methods) and drove 3D parameters using SyncLipMAE 's audio branch, the results were generally underwhelming—suggesting that additional geometry-aware supervision or models tailored to 3D priors are needed. (3) Factorization remains preliminary. Our decomposition into identity, vocal motion, and ambient motion is a first step; most of our downstream usage centers on the vocal-motion token. Systematically exploiting the other two factors (e.g., identity-aware editing, ambient-motion analysis/transfer) and strengthening disentanglement are promising directions for future work.

