# OpenReview forum: "SyncLipMAE: Contrastive Masked Pretraining for Audio–Visual Talking-Face Representations"
_ICLR.cc/2026/Conference — Submitted to ICLR 2026_

### Official Review · Reviewer_wRiN · 2025-10-27

**Soundness:** 1
**Presentation:** 2
**Contribution:** 3
**Rating:** 4
**Confidence:** 3

**Summary:**

The paper proposes a method to learn facial representations from talking-face video.
This is achieved through two main mechanisms:
(i) aligning visual representations to pretrained speech wav2vec embeddings, and
(ii) reconstructing the masked video frames in a masked autoencoding (MAE) fashion.
A distinctive feature of the approach is the attempt to factorize the face representation into three distinct components: identity, head movement, speech content.
The specialization of these components is primarily driven thorugh data augmentations (identity prompt shuffling, photometric perturbations, face-aware masking),
which encourage each of the components to retain specific aspects of the input.
The approach is evaluated on four downstream tasks: audio-visual synchronization, visual speech recognition, emotion/action recognition, and video dubbing.
However, the same datasets are used for both pretraining and downstream evaluation, limiting the evidence for transferability and complicating a comparison with existing baselines.

**Strengths:**

- The pretrained model, if released, could potentially be very useful to the community.
- The idea of factorizing the representation into specialized and disentangled components is conceptually appealing (although it's unclear whether this is really achieved; see weaknesses).
- The evaluation on a diverse set of downstream tasks is helpful.

**Weaknesses:**

- It's unclear whether the resulting representations are fully disentangled and whether they contain only the intended information. More importantly, is the proposed factorization even necessary, given that speech representation $\mathbf{z}^{\text voc}$ is used in all four downstream tasks?
- Pretraining and downstream datasets come from the same distribution of datasets. This is very important, since it limits the evidence for transferability and could give an advantage over the baselines, which is unclear whether they are retrained or finetuned on the same datasets.
- Many implementation details are omitted. How are the loss weights chosen? What are the exact baselines that the authors compare against (e.g., there are many AV-HuBERT checkpoints)?  What is the neighborhood size (self-similarity threshold) for the contrastive loss? Is the model trained on the full video or on segments?

**Questions:**

- See weaknesses; I would like the authors to address especially the second point, on the data overlap.
- Do the authors plan to release the pretrained model and the code upon acceptance?
- In section 4.2 L.290 the authors say that the linear probing the averaged $\mathbf{z}^\text{voc}$ and $\mathbf{z}^\text{amb}$ representations "cleanly separates speech-synchronized orofacial motion from audio-agnostic dynamics". Could the authors clarify how an average representation measures the disentanglement of the components? Shouldn't those representations have been probed separately to measure this?
- If I understand correctly, the model employs only static (per-frame) information; still it excels at audio-visual synchronization. I find this surprising: doesn't the model need to encode temporal information as well? For example, many of the synchronization models use 3d convolutions ([Synchformer](https://arxiv.org/pdf/2401.16423), [VocaLIST](https://arxiv.org/pdf/2204.02090), [SparseSync](https://arxiv.org/pdf/2210.07055))
- More superficial observations:
	- Tables start appearing early in the paper, but are only referenced at the end of the paper. This is distracting when reading.
	- Please use the parenthetical citation `\citep` instead of textual one `\cite`, where appropriate.
	- The top labels in Figure 1 should be shifted a bit more to the right (the MAE Decoder is not part of the "b) Downstream adaptation of SyncLipMAE")
	- The Sapiens-0.3B models is missing a citation.
- The authors might consider discussing the following papers as well:
	- Gao, Zheng, and Ioannis Patras. "Self-Supervised Facial Representation Learning with Facial Region Awareness." _CVPR_, 2024.
	- Zhang, Yuanhang, et al. "ES3: Evolving Self-Supervised Learning of Robust Audio-Visual Speech Representations." _CVPR_, 2024.
	- Yaman, Dogucan, et al. "Audio-Visual Speech Representation Expert for Enhanced Talking Face Video Generation and Evaluation." _arXiv_, 2024.

---

> ### Author Response · Authors · 2025-11-25
> **Response to Reviewer wRiN**
>
> **To Reviewer wRiN**
>
> We are deeply grateful to Reviewer wRiN for the detailed and highly insightful comments, which are very valuable for polishing and strengthening our work. Below we address the main concerns raised in your review.
>
> **1. Regarding pre-training/evaluation overlap and fairness.**
> Both Reviewers **UHD1** and **wRiN** raised a concern that SyncLipMAE may have been pre-trained on some datasets that are also used for evaluation (using their training partitions for pre-training and their test partitions for evaluation), whereas several baselines have never been trained on these datasets at all. This could give SyncLipMAE a distributional advantage that is not purely architectural.
>
> In response, we added **zero-shot** evaluations for both audio–visual synchronization (AV-Sync) and visual speech recognition (VSR), where neither SyncLipMAE nor the baselines are finetuned on the target datasets, so that their cross-dataset generalization can be compared on an equal footing.
>
> - **For audio–visual synchronization.**
>   We curate **100 zero-shot clips** from the open web spanning diverse talking-face scenarios (talks, news, singing, recitation, interviews), and **restrict to English** because most baselines are trained exclusively on English data (e.g., AV-HuBERT, Auto-AVSR). Results are summarized in Appendix Section C, Table 14 (reproduced below as Table a). Experiments show that, even in these generalization scenarios, our method still outperforms its counterparts.
>
>   **Table a. Zero-shot AV-Sync on 100 zero-shot clips.**
>
>   | Method                 | K = 1 Acc ↑ (%) | K = 1 Offset ↓ | K = 5 Acc ↑ (%) | K = 5 Offset ↓ | K = 15 Acc ↑ (%) | K = 15 Offset ↓ | R-prec (32) Top1 ↑ | R-prec (32) Top2 ↑ | R-prec (32) Top3 ↑ |
>   |------------------------|-----------------|----------------|-----------------|----------------|-------------------|------------------|---------------------|---------------------|---------------------|
>   | SyncNet-5              | 25.98           | 5.19           | 33.38           | 4.15           | 41.34             | 3.08             | 13.31               | 21.21               | 27.99               |
>   | VocaLiST-5             | 28.47           | 4.92           | 36.05           | 3.88           | 44.23             | 2.93             | 15.42               | 23.68               | 30.54               |
>   | StableSyncNet-16       | 36.20           | **3.19**       | 37.53           | 3.10           | 40.49             | 2.89             | 27.37               | 38.57               | 46.50               |
>   | **SyncLipMAE-1 (ours)**| **40.30**       | 3.78           | **50.71**       | **3.02**       | **61.67**         | **2.25**         | **33.66**           | **51.12**           | **61.33**           |
>
> - **For VSR.**
>   Because none of the compared models are trained on **CelebV-HQ** or **CelebV-Text**, we evaluate all methods **zero-shot** on the **test splits** of these datasets, after filtering out clips with **no speech** or **extremely noisy background audio**. We also report results on **RAVDESS**, a dataset that contains both spoken and sung utterances, to evaluate performance on talking and singing scenarios. The results are reported in Appendix Section C, Table 15 (and are also reproduced below in Table b for convenience).
>
>   **Table b. Zero-shot VSR (WER↓) on CelebV-HQ, CelebV-Text, and RAVDESS.**
>
>   | Method               | **CelebV-HQ** | **CelebV-Text** | **RAVDESS** |
>   |----------------------|:-------------:|:---------------:|:-----------:|
>   | AV-HuBERT            | 15.17         | 16.73           | 3.09        |
>   | Auto-AVSR            | 10.39         | 12.80           | 2.45        |
>   | **SyncLipMAE (ours)**| **10.05**     | **12.19**       | **2.42**    |
>
> **Conclusion.** These zero-shot results indicate that our quantitative gains primarily stem from **SyncLipMAE’s stronger cross-dataset generalization**, rather than from having pre-trained on the same datasets that are later used for evaluation. We have added the full zero-shot protocol and results to **Appendix Section C**.
>
> ---
>
> **2. On releasing code and pretrained models.**
> We confirm that we will release both the code and pretrained inference models for SyncLipMAE, regardless of the final acceptance decision. Our goal is for SyncLipMAE to serve as a useful resource for the community.
>
> ---

---

> > ### Author Response · Authors · 2025-11-25
> > **Following the above discussion**
> >
> > **3. On whether factorization is achieved and whether it is necessary.**
> > Both Reviewer DKgg and Reviewer wRiN emphasize that it is important to verify whether the proposed factorization is actually realized in the learned representation, and to justify why such a factorization is needed. We fully agree that this is crucial for the soundness of our claims and are grateful for this feedback.
> >
> > In the revised version, we add a dedicated subsection in main text **Section 3** titled *Prompt Token Factorization Analysis*, where we provide more direct evidence that the three prompts specialize into identity, vocal motion, and ambient motion:
> >
> > - **Reconstruction with prompt tokens from different sources.**
> >   We reconstruct masked portraits while sourcing the three prompts $\mathbf{z}^{\mathrm{id}}$, $\mathbf{z}^{\mathrm{voc}}$, and $\mathbf{z}^{\mathrm{amb}}$ from three frames with different identities and expressions (Fig. 4(a)). The reconstructions follow the ambient source in eye blinks (via $\mathbf{z}^{\mathrm{amb}}$) and the vocal source in mouth shape (via $\mathbf{z}^{\mathrm{voc}}$), while $\mathbf{z}^{\mathrm{id}}$ predominantly controls static appearance.
> >
> > - **Cross-attention patterns.**
> >   We aggregate cross-attention weights from all decoder blocks and visualize per-prompt attention maps (Fig. 4(b)): attention from $\mathbf{z}^{\mathrm{voc}}$ concentrates on the lower face (mainly the mouth), $\mathbf{z}^{\mathrm{amb}}$ spreads across the face with weak background response, and $\mathbf{z}^{\mathrm{id}}$ focuses more on background and non-mouth/eye regions.
> >
> >
> > Together, these analyses provide qualitative evidence that the three prompts form a factorized representation over identity, vocal motion, and ambient motion.
> >
> > In addition, we conduct an ablation study on the prompt-token composition (reported in Appendix B, Table 8, reproduced as Table c below) to assess the necessity and effectiveness of the three-token design. As introduced in the main paper, SyncLipMAE encodes each talking-face frame into a three-token representation $(\mathbf{z}^{\mathrm{id}}, \mathbf{z}^{\mathrm{amb}}, \mathbf{z}^{\mathrm{voc}})$. In this ablation, we systematically remove $\mathbf{z}^{\mathrm{id}}$, $\mathbf{z}^{\mathrm{amb}}$, or both, and evaluate how these variants affect the quality of the extracted $\mathbf{z}^{\mathrm{voc}}$ on the AV-Sync task. As shown in Experiment D (Table c), dropping any of the tokens consistently degrades the ability of $\mathbf{z}^{\mathrm{voc}}$ to match the audio tokens. We attribute this to the orthogonality loss in our objective, which explicitly encourages the three prompts to focus on complementary aspects of the talking-face frame, making the full three-token factorization strictly more effective than any reduced variant.
> >
> > **Table c. Ablations on prompt tokens (Hallo3 AV sync).**
> >
> > | Variant | Prompt tokens                      | K = 1 Acc ↑ (%) | K = 1 Offset ↓ | K = 5 Acc ↑ (%) | K = 5 Offset ↓ | K = 15 Acc ↑ (%) | K = 15 Offset ↓ | R-prec (32) Top1 ↑ | R-prec (32) Top2 ↑ | R-prec (32) Top3 ↑ |
> > |---------|------------------------------------|------------------|----------------|------------------|----------------|-------------------|------------------|---------------------|---------------------|---------------------|
> > | D1      | $z^{\mathrm{voc}}$                | 47.12           | 3.18           | 63.27           | 2.16           | 77.95            | 1.12             | 35.64               | 55.73               | 68.41               |
> > | D2      | $z^{\mathrm{id}}, z^{\mathrm{voc}}$   | 50.03           | 2.90           | 66.47           | 1.96           | 80.56            | 1.00             | 38.22               | 59.03               | 71.92               |
> > | D3      | $z^{\mathrm{amb}}, z^{\mathrm{voc}}$  | 51.41           | 2.74           | 67.39           | 1.82           | 81.33            | 0.97             | 39.26               | 60.37               | 72.81               |
> > | Ours    | $z^{\mathrm{id}}, z^{\mathrm{voc}}, z^{\mathrm{amb}}$ | **52.53** | **2.66** | **68.41** | **1.73** | **82.27** | **0.93** | **40.18** | **61.48** | **73.49** |
> >
> > ---
> >
> > **4. On missing implementation details.**
> > We apologize for the missing details in the original submission. In the revised version, we expand the *Experiments* section with additional implementation details and clearer dataset descriptions. For example, for AV-HuBERT we now explicitly state that we use the *Large + Self-Training* checkpoint pretrained on LRS3 and VoxCeleb2 (English) and finetuned on LRS3-433h.
> >
> > We also provide more detailed experimental settings in the appendix and upload the code as supplementary material with an expanded README, so that our training and evaluation pipelines can be more easily reproduced.
> >
> > ---

---

> > > ### Author Response · Authors · 2025-11-25
> > > **Following the above discussion part 2**
> > >
> > > **5. On using only per-frame information yet excelling at audio–visual synchronization.**
> > > You observe that our model employs only static (per-frame) visual tokens, yet performs very strongly on audio–visual synchronization. Indeed, we view this as one of the key findings of SyncLipMAE. Prior works typically use multi-frame windows as the minimal processing unit, e.g., 5 frames in SyncNet and VocaLiST, or 16 frames in LatentSync. In contrast, our experiments suggest that a *single* frame, paired with its corresponding short audio window (≈ 0.04 seconds), already carries sufficient information to support strong alignment.
> > >
> > > This is not completely without trade-offs: as shown in Appendix C, our single-frame configuration ($K = 1$) can be slightly weaker than some multi-frame baselines evaluted on some datasets, but as $K$ increases, the performance of SyncLipMAE improves more rapidly than that of models that use 5- or 16-frame windows as their minimal encoding unit. We clarify this behaviour and its implications for temporal modeling in the revised text.
> > >
> > > ---
> > >
> > > **6. On discussing additional related work.**
> > > We thank you for pointing us to three closely related works:
> > >
> > > - Gao and Patras, *“Self-Supervised Facial Representation Learning with Facial Region Awareness”* (CVPR 2024);
> > > - Zhang et al., *“ES3: Evolving Self-Supervised Learning of Robust Audio-Visual Speech Representations”* (CVPR 2024);
> > > - Yaman et al., *“Audio-Visual Speech Representation Expert for Enhanced Talking Face Video Generation and Evaluation”* (arXiv 2024).
> > >
> > > We have added citations and discussion of these works in the updated Related Work section, highlighting how they relate to our factorized, synchronization-aware pretraining. To the best of our knowledge, none of these methods have publicly released code or pretrained models, so we are currently unable to perform quantitative comparisons.
> > >
> > > ---
> > >
> > > **7. On writing, figures, and citation corrections.**
> > > We are grateful that you carefully pointed out several presentation issues; we have corrected them all in the revised version:
> > >
> > > - *“Tables start appearing early in the paper, but are only referenced at the end of the paper. This is distracting when reading.”*
> > >   We have moved each table as close as possible to its first reference in the text.
> > >
> > > - *“Please use the parenthetical citation `\citep` instead of textual one `\cite`, where appropriate.”*
> > >   We revisited all citations and replaced `\cite` with `\citep` where parenthetical citation is more appropriate.
> > >
> > > - *“The top labels in Figure 1 should be shifted a bit more to the right (the MAE Decoder is not part of the ‘(b) Downstream adaptation of SyncLipMAE’).”*
> > >   We have updated Figure 1 accordingly so that the panel labels and grouping more accurately reflect the architecture.
> > >
> > > - *“The Sapiens-0.3B model is missing a citation.”*
> > >   We apologize for this oversight and have added the missing citation for Sapiens-0.3B.
> > >
> > > ---
> > >
> > > **8. On correcting an overstatement regarding “clean separation”.**
> > > You correctly point out that the original wording in Section 4.2 (L.290) was inaccurate: we wrote that linear probing the averaged motion prompts “cleanly separates speech-synchronized orofacial motion from audio-agnostic dynamics.” This was a mistake in phrasing on our part, and we apologize for the confusion.
> > >
> > > In the revised text, we replace this sentence with a more faithful description of the experiment:
> > >
> > > > For each frame $t$, we form a motion descriptor by averaging the two motion prompts,
> > > > $$
> > > > u_t = \tfrac{1}{2}\big( \mathbf{z}^{\mathrm{voc}}_t + \mathbf{z}^{\mathrm{amb}}_t \big),
> > > > $$
> > > > feed $u_t$ into a single linear layer to obtain frame-level logits, and then average the logits over sampled frames before applying a softmax to obtain video-level predictions. The identity prompt is not used in this head, so the classifier operates purely on motion factors for emotion/action classification.
> > >
> > > We believe this wording more accurately reflects our original intent, namely that the classifier is applied purely to motion factors after explicitly removing identity information.

---

### Official Review · Reviewer_Vd5M · 2025-10-31

**Soundness:** 4
**Presentation:** 3
**Contribution:** 3
**Rating:** 8
**Confidence:** 5

**Summary:**

The paper presents an MAE (masked autoencoder) style self-supervised pretraining of an audio-visual talking face model. Each video frame is patchified and tokenized. Bypass 1 uniformly masks 75% of patches. Bypass 2 preserves the facial regions with a higher probability. The ViT-style encoder takes the visible patches and outputs prompt tokens consisting of the identity prompt, ambient-motion prompt, and vocal-motion prompt. In Bypass 1, the decoder takes the identity prompt along with the visible patch tokens to decode the video frame. In Bypass 2, the decoder takes the ambient-motion and vocal-motion prompts to decode the video frame. The audio feature per frame is a concatenation of per-layer features of a pretrained speech encoder (wav2vec 2.0) projected to the same dimension as the prompt tokens. The decoder is a sequence of transformer blocks taking visible patches from bypass 1 and mask tokens. Each block applies a self-attention followed by a cross-attention with identity and ambient tokens, and a conditioning token. The decoding process with shared decoder parameters performs two passes: a video-driven pass, where the conditioning token is the vocal-motion token, and an audio-driven pass, where he conditioning token is the audio feature. Both passes reconstruct the same video frame for symmetric supervision with visual and audio signals. The model is trained with the pixel reconstruction loss, CLIP-style audio-vocal contrastive alignment loss, and cross-covariance decorrelation penalty on the prompt tokens. The paper presents the applications of the model for audio-visual synchronization measurement, facial understanding, lip reading, and video dubbing with reference frames and mouth masks.

**Strengths:**

* Versatile model with multiple applications related to talking head with speech audio
* Well-thought-out design and training strategy supported by evaluations and ablation studies
* Honest disclosure of limitations with intriguing insights

**Weaknesses:**

* Only video dubbing visual results provided
  * I would like to see lip-reading results
* The paper could be argued as an application of MAE in a specific domain
  * Though I believe the paper has enough unique contributions (prompt token design and contrastive learning) on top of MAE

**Questions:**

Do authors have thoughts on evaluating the consistency of the identity prompt? Will the identity be preserved if the last frame of the previous segment is not provided in the video dubbing? What happens if the identity feature comes from face recognition models like ArcFace? Can identity prompts be used like face recognition?

---

> ### Author Response · Authors · 2025-11-25
> **Response to Reviewer Vd5M**
>
> ## To Reviewer Vd5M
>
> We sincerely thank Reviewer Vd5M for the comments on SyncLipMAE and the constructive concerns that help us further polish the work. We respond point by point below. If any part of our response is unclear or if we have misunderstood your questions, please do not hesitate to let us know.
>
> 1. **Regarding the demos.**
>    We have updated the supplementary material with more complete demos, including the VSR setting you are particularly interested in.
>
> 2. **Regarding “Will the identity be preserved if the last frame of the previous segment is not provided in the video dubbing?”**
>    In the video dubbing task, our goal is to synthesize the **masked lower half of the face** in the input video conditioned on the input audio. Consequently, if we do not provide a “reference image” that contains the full lower half of the face, the model can only freely hallucinate its appearance (e.g., whether there is a beard, the color and shape of the lips, details of the teeth, etc.).
>    Therefore, if neither a reference image nor the last frame of the previous segment is provided, the identity of the lower half of the face will not be preserved. If we provide only the reference image (without the last frame of the previous segment), the generated video can still preserve the identity, but when synthesizing long videos segment by segment, there may be visible temporal discontinuities at the boundaries between segments.
>
> 3. **Regarding “Can identity prompts be used like face recognition?” and how to evaluate “the consistency of the identity prompt”.**
>    Frankly, we are not experts in face recognition, but we view your question as a way to test whether the identity prompt is reliably extracted. We therefore conducted a simple experiment. On the HDTF dataset (which contains 349 distinct identities), we keep SyncLipMAE frozen, use its identity prompt as the feature representation, and train a linear classification head to predict the identity from each input face image. On a randomly selected set of 100 test images, the classifier achieves correct predictions for all samples.
>    In our understanding, this suggests that the identity prompts can indeed be used for face recognition and exhibit a certain degree of “consistency”. If this interpretation of your notion of consistency is inaccurate, we would be very grateful for your clarification.
>
> 4. **Regarding “What happens if the identity feature comes from face recognition models like ArcFace?”**
> Conceptually, this is feasible: both our identity prompt and features from face recognition models (e.g., ArcFace) encode slowly-varying, person-specific appearance. In principle, one could plug a frozen face recognition embedding into our architecture by projecting it to the prompt dimension and using it as the identity token for the decoder.
> However, the role and content of our identity prompt **differ from standard face recognition features** produced by models like ArcFace, and our use of the term “identity” is slightly broader than the strict notion in face recognition. Face recognition embeddings are explicitly trained for identity discrimination and have to be invariant to factors such as hairstyle, clothing, and background that do not help with deciding “who this person is”. In SyncLipMAE, by contrast, the identity token instead collects **all static, non-motion appearance information** — including facial shape, hair, clothing, and background — so that, **under our cross-covariance regularisation**, the **motion tokens are driven to capture appearance-agnostic dynamics only**. In the revised main text, Fig.~2(b) further supports this interpretation: by visualising the MAE-decoder attention maps, we observe that the **identity prompt predominantly attends to static regions** (background and relatively rigid facial areas), while assigning much **lower attention weights** to highly **dynamic regions** such as the mouth and eyes. We apologise if this terminology was potentially misleading in the current draft.

---

> > ### Comment · Reviewer_Vd5M · 2025-11-26
> >
> > I thank the authors for an insightful response. Learned identity prompt performing well in a face recognition task is another promising sign that the disentanglement is working as designed.
> >
> > And as the authors point out, the identity prompt in this context should indeed include the invariant features of the identity within the video, which differs from the objective of face recognition. In fact, I believe previous image/video generations using ArcFace (e.g. [Papantoniou et al. 2024]) suffered from this issue.
> >
> > I have no further questions. For now, I remain supportive of this paper.

---

### Official Review · Reviewer_DKgg · 2025-11-01

**Soundness:** 3
**Presentation:** 4
**Contribution:** 4
**Rating:** 8
**Confidence:** 5

**Summary:**

This is mainly a representational learning paper for 2D talking face videos. The authors present a novel method to factor 2D videos of talking faces into clean identity, vocal motion, and ambient motion latents.

The paper achieves this by using a contrastive-masked pre-training approach. For the identity token (Bypass 1), the paper uses a random uniform mask which hides a majority of the facial frames and forces the reconstruction of the face. For the motion tokens (Bypass 2), the paper utilizes a face-preserving mask combined with photometric jitter to effectively mask the identity from the encoder, forcing it to learn motion latents. Furthermore, a standard CLIP approach is used to improve the synchronisation between the per-frame audio tokens and vocal-motion tokens specifically, while penalizing cross-covariance between the token pairs.

The pre-trained model can then be used to generate tokens for four downstream tasks, namely audio-visual stream synchronization, facial understanding, visual speech recognition, and visual dubbing.

**Strengths:**

The paper presents both an original problem formulation of the face representation learning problem as well as a novel pre-training design. The two-bypass masking strategy is original and appears to work well on the 2D talking face setting.

The authors demonstrate the effectiveness of this learning strategy through adaptation of learnt latents into four orthogonal downstream tasks, achieving SOTA results across synchronization, understanding, VSR, and dubbing, with the zero-shot performance on AV sync being a strong evidence for generalization.

The paper is largely well-written and easy to follow.

**Weaknesses:**

The paper lacks qualitative or quantitative analysis to directly evaluate the factorization quality of the learnt latents. For example, it would be helpful to show that a change in the identity token while keeping the motion tokens static would change the person’s appearance but not the animation, nor the quality of the synchronization.
Furthermore, the paper lacks discussion on how this technique can be effectively generalized into similar problem domains, such as (most relevantly), 3D facial animation tasks. While the two-bypass approach is novel, it utilizes 2D techniques like image masking and photometric jitter, which do not have direct correspondence to 3D representations. Experimentation or discussion on more generalized approach would be greatly appreciated.

Finally, it seems like the training run utilized 128 GPUs for 15 days. The large amount of compute used on relatively low-resolution images is concerning, since it suggests that while the final representations learnt may be cleanly factorized, the formulation of the training task may suffer from poor sample efficiency which hinders convergence dynamics. An analysis of performance versus compute or data scale would be beneficial.

**Questions:**

Could the authors provide qualitative or quantitative analysis to directly support the claim of successful factorization, instead of purely relying on downstream tasks?
How can the core principle of using separate, differently-perturbed views to disentangle constants from motion be generalized to different problem tasks, like 3D facial animation (or others)?

The pre-training requires a substantial amount of compute on seemingly low-resolution images. Could the authors provide insight into the sample efficiency of the training objective? For instance, how does performance scale with the amount of data or compute, and is the complex objective the primary driver of the high computational cost? What happens when the objective is simplified?

Current works like VASA-1 also uses latent disentanglement yet formulates it within the losses in the end-to-end network. Could the authors discuss the primary advantages and tradeoffs for the motivation of more granular factorization through masked views here?

---

> ### Author Response · Authors · 2025-11-25
> **Response to Reviewer DKgg**
>
> ## To Reviewer DKgg
>
> We thank Reviewer DKgg for the professional and thoughtful comments on our work. Below we summarize the main concerns you raised and respond to them point by point.
>
> 1. **On whether factorization is achieved and whether it is necessary.**
> Both Reviewer **DKgg** and Reviewer **wRiN** emphasize that it is important to verify whether the proposed factorization is actually realized in the learned representation, and to justify why such a factorization is needed. We fully agree that this is crucial for the soundness of our claims and are grateful for this feedback.
> In the revised version, we add a dedicated subsection in main text **Section 3** titled *Prompt Token Factorization Analysis*, where we provide more direct evidence that the three prompts specialize into identity, vocal motion, and ambient motion:
>
> - **Reconstruction with prompt tokens from different sources.**
>   We reconstruct masked portraits while sourcing the three prompts $\mathbf{z}^{\mathrm{id}}$, $\mathbf{z}^{\mathrm{voc}}$, and $\mathbf{z}^{\mathrm{amb}}$ from three frames with different identities and expressions (Fig. 4(a)). The reconstructions follow the ambient source in eye blinks (via $\mathbf{z}^{\mathrm{amb}}$) and the vocal source in mouth shape (via $\mathbf{z}^{\mathrm{voc}}$), while $\mathbf{z}^{\mathrm{id}}$ predominantly controls static appearance.
>
> - **Cross-attention patterns.**
>   We aggregate cross-attention weights from all decoder blocks and visualize per-prompt attention maps (Fig. 4(b)): attention from $\mathbf{z}^{\mathrm{voc}}$ concentrates on the lower face (mainly the mouth), $\mathbf{z}^{\mathrm{amb}}$ spreads across the face with weak background response, and $\mathbf{z}^{\mathrm{id}}$ focuses more on background and non-mouth/eye regions.
>
>
> Together, these analyses provide qualitative evidence that the three prompts form a factorized representation over identity, vocal motion, and ambient motion.
>
> In addition, we conduct an ablation study on the prompt-token composition (reported in Appendix B, Table 8, reproduced as Table c below) to assess the necessity and effectiveness of the three-token design. As introduced in the main paper, SyncLipMAE encodes each talking-face frame into a three-token representation $(\mathbf{z}^{\mathrm{id}}, \mathbf{z}^{\mathrm{amb}}, \mathbf{z}^{\mathrm{voc}})$. In this ablation, we systematically remove $\mathbf{z}^{\mathrm{id}}$, $\mathbf{z}^{\mathrm{amb}}$, or both, and evaluate how these variants affect the quality of the extracted $\mathbf{z}^{\mathrm{voc}}$ on the AV-Sync task. As shown in Experiment D (Table c), dropping any of the tokens consistently degrades the ability of $\mathbf{z}^{\mathrm{voc}}$ to match the audio tokens. We attribute this to the orthogonality loss in our objective, which explicitly encourages the three prompts to focus on complementary aspects of the talking-face frame, making the full three-token factorization strictly more effective than any reduced variant.
>
>
>
>
> **Table c. Ablations on prompt tokens (Hallo3 AV sync).**
>
> | Variant | Prompt tokens                      | K = 1 Acc ↑ (%) | K = 1 Offset ↓ | K = 5 Acc ↑ (%) | K = 5 Offset ↓ | K = 15 Acc ↑ (%) | K = 15 Offset ↓ | R-prec (32) Top1 ↑ | R-prec (32) Top2 ↑ | R-prec (32) Top3 ↑ |
> |---------|------------------------------------|------------------|----------------|------------------|----------------|-------------------|------------------|---------------------|---------------------|---------------------|
> | D1      | $z^{\mathrm{voc}}$                | 47.12           | 3.18           | 63.27           | 2.16           | 77.95            | 1.12             | 35.64               | 55.73               | 68.41               |
> | D2      | $z^{\mathrm{id}}, z^{\mathrm{voc}}$   | 50.03           | 2.90           | 66.47           | 1.96           | 80.56            | 1.00             | 38.22               | 59.03               | 71.92               |
> | D3      | $z^{\mathrm{amb}}, z^{\mathrm{voc}}$  | 51.41           | 2.74           | 67.39           | 1.82           | 81.33            | 0.97             | 39.26               | 60.37               | 72.81               |
> | Ours    | $z^{\mathrm{id}}, z^{\mathrm{voc}}, z^{\mathrm{amb}}$ | **52.53** | **2.66** | **68.41** | **1.73** | **82.27** | **0.93** | **40.18** | **61.48** | **73.49** |

---

> ### Author Response · Authors · 2025-11-25
> **Following the above discussion**
>
> 2. **On extending to 3D face animation and related domains.**
> We discuss this from two angles.
>
> - **Can 3D face animation serve as an additional downstream adaptation task for SyncLipMAE?**
> We perform an exploratory experiment where SyncLipMAE is used to drive a 3D talking-face generator (based on 3DMM parameters, with a generator architecture the same as UniTalker). Specifically, we consider two driving modes: (i) using the audio features extracted by SyncLipMAE, and (ii) using the visual vocal-motion prompt $\mathbf{z}^{\mathrm{voc}}$, which is aligned with the corresponding audio, as the driving signal. As shown in the last page of the updated demo, both modes can produce reasonable 3D face animations that are aligned with either the audio or the source video. Because this is only a preliminary, proof-of-concept experiment, we did not conduct a detailed quantitative evaluation; however, despite some jitter and occasional lip–video mismatches in the zero-shot video-driven setting, these results suggest that using SyncLipMAE as a driving representation for 3D faces is a promising direction for our future work.
>
> - **Whether the pretraining strategy (factorization + audio contrast + MAE) is applicable to 3D faces.**
> In our current formulation, identity, ambient motion, and vocal motion are extracted from 2D face images. We use an orthogonality loss to encourage each prompt token to capture different aspects of the frame, and an audio–visual contrastive objective to make the vocal-motion prompt $\mathbf{z}^{\mathrm{voc}}$ strongly correlated with speech. We believe these two ingredients—the explicit disentangling pressure between prompts and the audio-aligned vocal-motion code—are also very natural for 3D faces, in particular for 3DMM-based representations (we are less familiar with other 3D face parameterizations).
>
> However, we do not yet have a clear answer on whether the MAE-style masking-and-reconstruction strategy we use for 2D images is equally suitable for 3DMM sequences. MAE operates by dropping spatial patches from an image, whereas 3DMM trajectories are essentially 1D temporal sequences of parameters rather than spatial grids. Extending our masked-view design and factorization objective to 3DMM therefore seems promising, but likely requires rethinking the masking scheme so that it better matches this temporal-parameter setting.
>
> ---
>
> 3. **On training efficiency and the role of the complex objective.**
> The main training cost of SyncLipMAE comes from the intrinsic difficulty of learning a reliable correspondence between lip motion and audio, rather than from the MAE reconstruction term itself. In our setting, this difficulty arises mainly from two factors:
>
> - **Intrinsic ambiguity of visual speech (viseme–phoneme mismatch).**
>   Visual speech is inherently ambiguous: many phonemes share almost indistinguishable lip shapes and are grouped into the same viseme class (e.g., /p/, /b/, and /m/ all look like a closed mouth). This many-to-one mapping from phonemes to visemes makes it harder for the contrastive loss to learn a precise alignment between audio tokens and visual vocal-motion tokens.
>
> - **Non-speech and mixed audio in general scenarios.**
>   In general in-the-wild settings, the audio track is often not “pure” speech. For example, singing typically comes with background music, and TV or stage programs may include audience applause, laughter, or other sound effects. These additional sources make the audio–visual correspondence noisier and further increase the difficulty of contrastive learning.
>
> Taken together, our experiments indicate that the dominant training cost comes from the audio–visual contrastive learning needed to handle these ambiguities and noisy conditions, rather than from the other loss terms (e.g., the MAE-style reconstruction loss).

---

> > ### Author Response · Authors · 2025-11-25
> > **Following the above discussion part 2**
> >
> > **4. On the difference between VASA-style disentanglement and SyncLipMAE.**
> > We appreciate Reviewer DKgg’s question about how our factorization compares to the “VASA-1 style” of disentanglement; this aligns closely with the design choices we considered when developing SyncLipMAE.
> >
> > Very roughly summarizing our understanding of the VASA-style pipeline (e.g., One-shot free-view neural talking-head synthesis, MegaPortrait, LivePortrait, VASA-1):
> >
> > - **Step 1.** A 3D canonical head representation (e.g., a voxel field or implicit 3D feature volume) and a set of canonical 3D keypoints are first estimated from a single reference image.
> > - **Step 2.** For each target frame, 3D keypoints encode head pose and facial expression. Local affine transforms of these keypoints are computed relative to the canonical configuration.
> > - **Step 3.** These sparse keypoint transforms are then propagated to a dense deformation of the 3D head volume, yielding a pose- and expression-specific 3D field per frame.
> > - **Step 4.** Finally, this 3D volume is projected to 2D and fed through a generator (e.g., a SPADE-style network) to obtain the rendered talking-face frame.
> >
> > In other words, “face motion” in VASA-style methods is represented primarily in a **3D keypoint / 3D head space**, and the disentanglement between identity and motion relies on: (i) a canonical 3D head volume extracted from the reference image, and (ii) 3D keypoint trajectories that drive deformations of this volume.
> >
> > In our experiments with LivePortrait, MegaPortrait, and related methods, we observed a **fundamental limitation** of this representation: the 3D keypoint–based motion remains partially entangled with **head shape and facial layout**, and it relies heavily on a strong 3D head voxel (or parametric) extractor. In many VASA-style pipelines, this entanglement is further reinforced by supervising the motion branch with 2D/3D facial landmarks, whose positions are themselves strongly dependent on the underlying head shape and facial geometry. This coupling was one of the main motivations for our design in SyncLipMAE, where we **avoid using 3D keypoint–based motion representations** and instead factorize motion directly in a **2D latent space** via three prompts (identity, ambient motion, vocal motion). In addition, VASA-style works typically focus on high-fidelity talking-face generation and **do not explicitly align motion with audio**; any correspondence between audio and motion is learned implicitly via the generative objective. By contrast, SyncLipMAE uses:
> >
> > - an **orthogonality loss** to encourage each prompt token to capture different aspects of the frame (identity vs. ambient vs. vocal motion), and
> > - an explicit **audio–visual contrastive objective** to make the vocal-motion prompt $\mathbf{z}^{\mathrm{voc}}$ tightly aligned with speech.
> >
> > This explicit audio–motion interface is what allows SyncLipMAE to support a broader set of downstream tasks (AV sync, VSR, understanding, dubbing), rather than being limited to talking-face generation.
> >
> > That said, we also acknowledge that VASA-style representations have important strengths. In particular, their explicit 3D formulation is **highly efficient for generating 2D talking-face videos**, because it provides a **direct and well-structured 3D-to-2D mapping** pipeline. This is one of the main reasons why, to the best of our knowledge, most recent real-time talking-face systems (e.g., ChatAnyOne) adopt VASA-like 3D face motion representations.
> >
> > To make the contrast more concrete, we summarize the key conceptual differences in Table a:
> >
> > **Table a. Conceptual comparison between VASA-style methods and SyncLipMAE.**
> >
> > | Method family                                | Motion space                                   | Explicit audio–motion alignment? | Requires 3D head voxel / parametric extractor? |
> > |---------------------------------------------|-----------------------------------------------|----------------------------------|-----------------------------------------------|
> > | VASA-style                                  | 3D keypoints / 3D head motion                 | ✗                                | ✓                                             |
> > | SyncLipMAE                                  | 2D latent prompts $(\mathbf{z}^{\mathrm{id}}, \mathbf{z}^{\mathrm{amb}}, \mathbf{z}^{\mathrm{voc}})$ | ✓                                | ✗                                             |

---

### Official Review · Reviewer_UHD1 · 2025-11-01

**Soundness:** 3
**Presentation:** 2
**Contribution:** 3
**Rating:** 2
**Confidence:** 4

**Summary:**

This paper presents a self-supervised approach based on masked autoencoders which learns audio and visual features from unlabelled videos. Then the pre-trained model is used on 4 different applications: visual speech recognition, audio-visual synchronisation, visual dubbing and emotion classification and achieving state-of-the-art results.

**Strengths:**

- The idea of pretraining an audio-visual model which can be used for multiple audio-visual downstream tasks is interesting.

- The proposed approach seems to achieve state-of-the-art results.

**Weaknesses:**

- The paper is hard to follow, although the general idea is clear, the fact that so many different things are presented leaves the reader a bit confused. For example, how the models are trained for each downstream task is briefly presented (probably due to lack of space) and the process is not entirely clear. Adding additional figures showing which pre-trained models are used for each task and how they are combined with the syncLipMAE would be very informative. In addition, Figure 1 is a bit confusing. It's hard for the reader to fully understand how the approach works.

- Details on the datasets used are missing. It would good if the number of subjects, number of videos and total duration of each dataset are shown including how the data are divided into (raining/validation/test sets. Also, a table showing the datasets used in each downstream task would improve the clarity of the dataset section.

- Non-standard datasets are used for evaluating the models for VSR and AV synchronisation. For example, LRS2 and LRS3 are the standard benchmarks used for VSR. Why not using them?

- The model has been pretrained on Hallo3, CelebV-HQ, CelebV-Text, MEAD, VFHQ, HDTF and RAVDESS. The same datasets are used for evaluation, e.g., facial understanding is evaluated on CelebV-HQ, AV synchronisation and visual dubbing are evaluated on Hallo3 and VFHQ and finally VSR is evaluated on HDTF, VFHQ, and Hallo3. As a consequence, the comparison with other models is not fair as they have been trained on different datasets (and the proposed has an advantage).

- The authors report SOTA results in the paper, however the visual dubbing results presented in the supplementary material do not seem SOTA.

**Questions:**

Please see above.

---

> ### Author Response · Authors · 2025-11-25
> **Response to Reviewer UHD1**
>
> ## To Reviewer UHD1
>
> We thank Reviewer UHD1 for the careful reading of our paper and for the constructive suggestions on improving clarity, dataset documentation, and evaluation protocols. Below, we address each of the concerns raised by Reviewer UHD1 in turn.
>
> **Regarding pre-training/evaluation overlap and fairness.**
> Both Reviewers **UHD1** and **wRiN** raised a concern that SyncLipMAE may have been pre-trained on some datasets that are also used for evaluation (e.g., using their training partitions for pre-training and their test partitions for evaluation), whereas several baselines have never been trained on these datasets at all. This could give SyncLipMAE a distributional advantage that is not purely architectural.
>
> In response, we added **zero-shot** evaluations for both audio–visual synchronization (AV-Sync) and visual speech recognition (VSR), where neither SyncLipMAE nor the baselines are finetuned on the target datasets, so that their cross-dataset generalization can be compared on an equal footing.
>
> - **For audio–visual synchronization.**
>   We curate **100 zero-shot clips** from the open web spanning diverse talking-face scenarios (talks, news, singing, recitation, interviews), and **restrict to English** because most baselines are trained exclusively on English data (e.g., AV-HuBERT, Auto-AVSR). Results are summarized in Appendix Section C, Table 14 (reproduced below as Table a). Experiments show that, even in these generalization scenarios, our method still outperforms its counterparts.
>
>   **Table a. Zero-shot AV-Sync on 100 zero-shot clips.**
>
> | Method                  | K = 1 Acc ↑ (%) | K = 1 Offset ↓ | K = 5 Acc ↑ (%) | K = 5 Offset ↓ | K = 15 Acc ↑ (%) | K = 15 Offset ↓ | R-prec (32) Top1 ↑ | R-prec (32) Top2 ↑ | R-prec (32) Top3 ↑ |
> |-------------------------|-----------------|----------------|-----------------|----------------|-------------------|------------------|---------------------|---------------------|---------------------|
> | SyncNet-5               | 25.98           | 5.19           | 33.38           | 4.15           | 41.34             | 3.08             | 13.31               | 21.21               | 27.99               |
> | VocaLiST-5              | 25.77           | 5.10           | 29.82           | 4.51           | 35.46             | 3.73             | 13.28               | 21.44               | 27.51               |
> | StableSyncNet-16        | 36.20           | **3.19**       | 37.53           | 3.10           | 40.49             | 2.89             | 27.37               | 38.57               | 46.50               |
> | **SyncLipMAE-1 (ours)** | **40.30**       | 3.78           | **50.71**       | **3.02**       | **61.67**         | **2.25**         | **33.66**           | **51.12**           | **61.33**           |
>
> - **For VSR.**
>   Because none of the compared models are trained on **CelebV-HQ** or **CelebV-Text**, we evaluate all methods **zero-shot** on the **test splits** of these datasets, after filtering out clips with **no speech** or **extremely noisy background audio**. We also report results on **RAVDESS**, a dataset that contains both spoken and sung utterances, to evaluate performance on talking and singing scenarios. The results are reported in Appendix Section C, Table 15 (and are also reproduced below in Table b for convenience).
>
>   **Table b. Zero-shot VSR (WER↓) on CelebV-HQ, CelebV-Text, and RAVDESS.**
>
>   | Method               | **CelebV-HQ** | **CelebV-Text** | **RAVDESS** |
>   |----------------------|:-------------:|:---------------:|:-----------:|
>   | AV-HuBERT            | 15.17         | 16.73           | 3.09        |
>   | Auto-AVSR            | 10.39         | 12.80           | 2.45        |
>   | **SyncLipMAE (ours)**| **10.05**     | **12.19**       | **2.42**    |
>
> **Conclusion.** These zero-shot results indicate that our quantitative gains primarily stem from **SyncLipMAE’s stronger cross-dataset generalization**, rather than from having pre-trained on the same datasets that are later used for evaluation.

---

> > ### Author Response · Authors · 2025-11-25
> > **Following the above discussion**
> >
> > **Regarding clearer descriptions of downstream tasks.**
> > We have expanded Section 4 in the main text to more clearly explain how SyncLipMAE is adapted to each downstream task. In particular, we have added a new Table 2  in the main text(reproduced below as Table c) that summarizes, for every task, the downstream head used and which prompt tokens from SyncLipMAE (identity, vocal motion, ambient motion) are fed into that head. We hope this additional structure makes it easier to understand how SyncLipMAE works.
> >
> > **Table c. Used heads and prompt tokens.**
> >
> > | Task                 | Downstream head          | Tokens                                   |
> > |----------------------|--------------------------|------------------------------------------|
> > | AV synchronisation   | Similarity (no head)     | $\mathbf{A}_t$, $\mathbf{z}^{\mathrm{voc}}_t$ |
> > | Facial understanding | Linear classifier        | $\mathbf{z}^{\mathrm{voc}}_t$, $\mathbf{z}^{\mathrm{amb}}_t$ |
> > | Lip reading (VSR)    | Conformer+Transformer    | $\mathbf{z}^{\mathrm{voc}}_t$            |
> > | Video dubbing        | WanVACE+AudioPack        | $\mathbf{A}_t$ / $\mathbf{z}^{\mathrm{voc}}_t$ |
> >
> > **Regarding demos for dubbing and VSR.**
> > We have updated the demo package to include more comprehensive visual dubbing and VSR examples. The new demos contain both audio-driven and video-driven dubbing cases, as well as side-by-side comparisons with LatentSync for representative scenarios, together with qualitative VSR samples. We hope these additional results provide a more convincing and transparent picture of the model’s behaviour.
> >
> > **Regarding dataset details.**
> > We apologize for the lack of detail in the original submission. In the revised version, we augment the *Experiments* section with a dedicated paragraph on “SyncLipMAE training data,” where we now explicitly describe the pre-training data preparation:
> >
> > > We pretrain SyncLipMAE on Hallo3, CelebV-HQ, CelebV-Text, MEAD, VFHQ, HDTF, and RAVDESS. Audio is converted to mono 16 kHz and videos are resampled to 25 fps. For VFHQ and MEAD we follow the official evaluation protocols, using the released test partitions for evaluation and the remaining clips for pretraining, and additionally sampling 100 training clips for validation. For Hallo3, CelebV-HQ, CelebV-Text, HDTF, and RAVDESS, which lack standardized splits in our setting, we randomly sample 100 clips for validation and 100 clips for testing, and use the rest for pretraining. All videos are decoded, uniformly resized, and center-cropped to 512×512, ensuring that the full face remains.
> >
> > Due to space limits in the main text, we further provide a separate dataset-description section in Appendix A, where we summarize the statistics and usage of each dataset.
> >
> > **Regarding not using LRS2/LRS3.**
> > The main reason we did not include LRS2/LRS3 in our experiments is a mismatch in video resolution and crop strategy. SyncLipMAE is trained on **512×512** frames that contain the **entire head** (including neck, forehead and hair), whereas the publicly released LRS2/LRS3 provide only **224×224** low-resolution face crops of the speaker, **without the full head**. Directly mixing these crops with our training data would require redesigning the architecture and data pipeline. We contacted the LRS2 authors to inquire about access to the original high-resolution videos but have not received a response, and, to the best of our knowledge, the current **LRS3 project pages no longer provide download links from any official host**. (https://mmai.io/datasets/lip_reading/#download shows "Downloads are no longer available from this website.")

---

### Author Response · Authors · 2025-11-25
**Overall Author Response, Revisions, and Summary of Reviews**

**To all reviewers:**

We first sincerely thank four reviewers for their careful reading of our submission and for the many insightful comments and suggestions that help us polish and better position this work.

We provide a brief summary of our work:
Our work introduces SyncLipMAE, a self-supervised pretraining framework for talking-face video that learns synchronization-aware and transferable facial dynamics from unlabeled audio–visual streams. Concretely, SyncLipMAE couples masked visual modeling with cross-modal contrastive alignment, and uses three per-frame prompt tokens that explicitly encode the key factors of a talking-face frame: identity, vocal motion (speech-synchronized facial dynamics), and ambient motion (audio-agnostic movements such as blinks and head pose). The contrastive objective treats time-aligned vocal-motion and audio tokens as positives and misaligned pairs as negatives, driving both modalities into a shared embedding space and enabling token-level audio–visual synchronization. After pretraining, we use the extracted audio tokens together with the prompt tokens to support four categories of downstream adaptations: (i) audio–visual stream synchronization, (ii) facial emotion and head/face action recognition, (iii) visual speech recognition, and (iv) visual dubbing, where we, to our knowledge for the first time, enable indistinguishable audio- or video-driven control in a unified framework.

**Summary of recognized strengths.**

Across the four reviews, several positive aspects are highlighted (Reviews UHD1, DKgg, Vd5M, wRiN):

- **(i)** A versatile audio–visual talking-head model that can support multiple downstream tasks within a single pretrained backbone, with evaluation on a diverse set of tasks (AV sync, VSR, facial understanding, dubbing) considered helpful. (Reviewers UHD1, DKgg, Vd5M, wRiN)

- **(ii)** A conceptually appealing factorization of talking-face representation into specialized components (identity, vocal motion, ambient motion), realized via an original two-bypass masking strategy, prompt-token design, and contrastive alignment. (Reviewers DKgg, Vd5M, wRiN)

- **(iii)** A well-thought-out design and training strategy, largely clear and easy-to-follow writing with honest discussion of limitations, and a pretrained model that could be broadly useful to the community if released. (Reviewers DKgg, Vd5M, wRiN)

- **(iv)** The pretrained model could be broadly useful to the community if released. (Reviewers wRiN)

**Summary of main concerns (framed as opportunities for improvement).**

The reviewers also raise several important points that have guided our revisions:

- (i) Missing implementation and data-usage details, including some training hyperparameters (e.g., loss weights) and clearer specification of which datasets are used for each downstream task (Reviewers UHD1, wRiN).
- (ii) The need for more direct analyses of the factorization quality of the learned latents (identity, vocal motion, ambient motion), rather than inferring this solely from downstream task performance (Reviewers DKgg, wRiN).
- (iii) Clarification of whether differences in training data introduce unfair comparisons, given that some prior methods may not have been trained on the training splits corresponding to our evaluation datasets (Reviewers UHD1, wRiN).
- (iv) More convincing demos, especially for visual dubbing and VSR, to better support the quantitative results (Reviewers UHD1, Vd5M).
- (v) A discussion of how the core ideas and architecture might transfer to related domains, such as 3D face animation and face recognition (Reviewers DKgg, Vd5M).

**Revisions and additional material.**

In the revised version submitted with this response, we have updated both the main text and the appendix to address the concerns raised by the reviewers, and we have also uploaded a more complete demo, including additional audio-driven and video-driven visual dubbing results as well as VSR samples. We next respond to each reviewer in turn, addressing all reported weaknesses and questions point by point, and we once again thank the reviewers for their careful and professional effort.

---

### Meta-Review · Area_Chair_nVec · 2026-01-06

**Summary:**

In the initial reviews, 2 reviewers recommend acceptance and 2 recommend rejection.

Reviewer UHD1 recommended rejection. They found the paper difficult to read and missing details on the method and data (though they acknowledge that some of this may be due to the lack of space). They complain about nonstandard benchmarking for VSR, specifically the missing results from LRS2 and LRS3. They also felt that the qualitative results for the visual dubbing were not competitive with SOTA. Along with wRiN, they complain that the same datasets were used for pretraining and evaluation. The rebuttal addresses this by proposing a new 100-video dataset for synchronization, updating the qualitative results with comparison to another method, and explaining why they didn't use LRS2/LRS3 (due to the limited resolution and the fact images don't provide the full talking head, with full videos not readily available). Reviewer DKgg appreciates the pretraining formulation, which creates special tokens that capture identity separately from motion, and they appreciate the results. However, they (along with wRiN) raise concerns about the lack of evidence for the model learning the claimed factorization, and question the sample efficiency of the model. To address these concerns, the authors evaluate the reconstruction behavior when giving the model latent vectors sourced from different examples. Reviewer Vd5M is in favor of acceptance due to the strength of the formulation but raises concerns about novelty. The authors update these demos and explain the differences in the identity features from other feature learning approaches approaches. The reviewer replied to say that they remain supportive of the paper. Reviewer wRiN is in favor of rejection. Like others, they raise concerns about the evidence for disentanglement, the train/test overlap, and missing implementation details. However, they praise the usefulness of the released system.

The AC agrees with reviewers wRiN and Vd5M that the paper presents an interesting and useful system that will be useful for other researchers. However, the AC agrees with wRiN and UHD1 that the nonstandard evaluations and training/test split are serious issues, especially for the synchronization evaluation, which is commonly evaluated zero shot. While the rebuttal partly addresses this by proposing a new sychronization dataset and describing how differences in image resolution and composition make this challenging, the lack of apples-to-apples comparisons to existing methods remains an issue. The rebuttal partly address the shortcomings in the evaluation of the factorization. On balance, the AC weighs the evaluation issues over the quality of the results, and feels that the paper requires revision before it can be accepted.

**Reviewer Concerns:**

As described in the summary, the rebuttal partly addresses UHD1's concerns by creating a new evaluation for the synchronization task, but it does not fully address their concerns about apples-to-apples comparisons for VSR (see above). The rebuttal helps address the factorization questions raised by DKgg. Reviewer Vd5M found the rebuttal to address their concerns. Reviewer wRiN's issues were similar to those of UHD1, therfore (like UHD1) their complaints are likely partly addressed by the rebuttal.

**Reviewer Scores:**

After the rebuttal, it is possible that UHD1 and wRiN would raise their scores due to the new experiments. However, they both raise fairly serious issues with how experiments were conducted, so it is unclear whether this would be sufficient. The other two reviewers would likely maintain their accept scores (especially Vd5M, which said the rebuttal addressed their issues).

---

### Decision · Program_Chairs · 2026-01-26

Reject